# `CauScale`: Neural Causal Discovery at Scale

**Bo Peng** [1 2 3]   **Sirui Chen** [2 4]   **Jiaguo Tian** [1]   **Yu Qiao** [2 3]   **Chaochao Lu** [2]

## Abstract

Causal discovery is essential for advancing data-driven fields such as scientific AI and data analysis, yet existing approaches face significant time- and space-efficiency bottlenecks when scaling to large graphs. To address this challenge, we present `CauScale`, a neural architecture designed for efficient causal discovery that scales inference to graphs with up to 1000 nodes. `CauScale` improves time efficiency via a reduction unit that compresses data embeddings and improves space efficiency by adopting tied attention weights to avoid maintaining axis-specific attention maps. To keep high causal discovery accuracy, `CauScale` adopts a two-stream design: a data stream extracts relational evidence from high-dimensional observations, while a graph stream integrates statistical graph priors and preserves key structural signals. `CauScale` successfully scales to 500-node graphs during training, where prior work fails due to space limitations. Across testing data with varying graph scales and causal mechanisms, `CauScale` achieves 99.6% mAP on in-distribution data and 84.4% on out-of-distribution data, while delivering $4\times$–$13,000\times$ inference speedups over prior methods. Our project page is at https://github.com/OpenCausaLab/CauScale.

## 1. Introduction

Causal discovery focus on uncovering causal relationships and mechanisms from observational data (Spirtes et al., 2000; Pearl, 2009; Glymour et al., 2019). A central component of causal discovery is causal structure learning, which identifies the underlying structural causal models (SCMs) and learns directed acyclic graphs (DAGs) where edges rep-

resent direct causal relationships between variables (Peters et al., 2017). Causal relationship inference is an important problem across many fields including bioinformatics (Sachs et al., 2005; Zhang et al., 2013), epidemiology (Vandenbroucke et al., 2016), and economics (Hicks, 1980).

As data grow increasingly complex, discovering causal relationships from massive datasets has become an urgent challenge. However, existing causal discovery algorithms face major time- and space-efficiency bottlenecks, particularly when scaling to large graphs. Constraint-based algorithms (e.g., PC and FCI (Spirtes et al., 2000)) can become time-prohibitive because they rely on large numbers of conditional-independence tests, whose count grows exponentially in the worst case. In contrast, score-based methods such as NOTEARS (Zheng et al., 2018) and RL-BIC (Zhu et al., 2019) avoid explicit combinatorial search but typically require solving a fresh continuous optimization problem for each dataset, which remains computationally expensive at scale. To reduce runtime, AVICI (Lorch et al., 2022) amortizes causal discovery by pretraining a supervised model on simulated data and performing zero-shot graph prediction at test time. However, its attention mechanism scales unfavorably with the number of variables, often leading to substantial memory pressure on large graphs.

To overcome these time- and space-efficiency bottlenecks, we propose `CauScale`, an efficient neural architecture for amortized causal discovery. Overall, `CauScale` adopts a two-stream design with a data stream and a graph stream. For time efficiency, we introduce a *reduction unit* that compresses the data embeddings during network processing. For space efficiency, we adopt tied attention weights (Rao et al., 2021) in both streams: sharing attention weights across axis avoids maintaining axis-specific attention maps and substantially reduces the memory footprint of attention. To improve efficiency without sacrificing discovery quality, we further design a *data-graph block* that (i) injects graph-prior information and (ii) mitigates information loss from data reduction. Specifically, it distills relational evidence from high-dimensional data into a graph message to guide representation learning in the graph stream. Moreover, it fuses the two streams by injecting the data stream into the graph embedding before reduction, so that the model preserves key relational signals and alleviates information loss.

---

[1]Shanghai Jiao Tong University [2]Shanghai Artificial Intelligence Laboratory [3]Shanghai Innovation Institute [4]Tongji University. Correspondence to: Chaochao Lu <luchaochao@pjlab.org.cn>.

*Proceedings of the 43rd International Conference on Machine Learning*, Seoul, South Korea. PMLR 306, 2026. Copyright 2026 by the author(s).

We conduct extensive experiments on synthetic and single-cell expression datasets with varying sizes and causal structures. The results demonstrate that CauScale achieves superior accuracy with markedly improved efficiency. Specifically, CauScale achieves an mAP of 99.6% on in-distribution data and 84.4% on out-of-distribution (OOD) data. It stand out as the fastest method, outperforming previous approaches by $4\times$ to $13,000\times$. Furthermore, during training, CauScale scales successfully to 500-node graphs, a setting where AVICI fails due to limited memory and excessive space costs.

In summary, our contributions are:

- We present one of the first studies on pre-training neural networks for efficient causal discovery at scale, offering a scalable step toward uncovering causal relations from increasingly complex data.

- We introduce CauScale, a neural architecture that jointly improves time and memory efficiency with high causal discovery accuracy.

- We conduct comprehensive experiments to validate the effectiveness of CauScale across varying graph scales and causal mechanisms, demonstrating that CauScale improves both efficiency and causal discovery performance.

## 2. Related Work

Existing causal structure learning methods can be broadly categorized into *non-amortized* and *amortized (zero-shot)* approaches, relying on whether inference on a new dataset requires solving a dataset-specific optimization problem.

**Non-amortized causal discovery.** Non-amortized methods perform causal discovery by solving an optimization or search problem independently for each dataset, leading to high computational cost and limited scalability. 1) **Constraint-based algorithms**, such as PC and FCI (Spirtes et al., 2000), infer graph structures via conditional independence tests but suffer from exponential complexity as the number of variables grows. 2) **Score-based algorithms** optimize a predefined score over the space of graph structures (Tsamardinos et al., 2006; Goudet et al., 2017). Classical approaches rely on greedy combinatorial search, including GES (Chickering, 2002) and GIES (Hauser & Bühlmann, 2012). To improve scalability, recent work reformulates causal discovery as continuous optimization with differentiable acyclicity constraints (Zheng et al., 2018; Lachapelle et al., 2019; Ke et al., 2019; Zhu et al., 2019; Brouillard et al., 2020). NOTEARS (Zheng et al., 2018) introduces a smooth acyclicity constraint, while SDCD (Nazaret et al.) further improves stability via spectral constraints and staged optimization. 3) **Functional Causal Model (FCM) based**

**methods** exploit asymmetries in the data-generating process for identifiability. Early approaches such as LiNGAM (Shimizu et al., 2006) rely on Independent Component Analysis (ICA) (Hyvärinen et al., 2001), whereas recent methods integrate deep generative models. For example, DiffAN (Sanchez et al., 2023) frames causal discovery as topological sorting using diffusion-based score estimators. 4) **Granger causality based methods** infer directional relationships from temporal precedence, and are therefore designed for *sequential* rather than i.i.d. data. Classical approaches exploit spectral decompositions of multivariate time series (Gallagher et al., 2017), while neural extensions capture nonlinear dependencies via autoregressive networks (Tank et al., 2021). Recent work further leverages generative factor models to hypothesize dynamic causal graphs from observed time series (Brown & Carlson, 2025). Despite their diversity, non-amortized methods require dataset-specific optimization, making them computationally expensive and unsuitable for large-scale or real-time inference.

**Amortized (zero-shot) causal discovery.** Amortized approaches aim to eliminate dataset-specific optimization by learning a shared inference model that maps datasets directly to causal graphs, enabling zero-shot inference on unseen data (Lorch et al., 2022; Ke et al., 2023; Wu et al., 2025; Dhir et al., 2025). Lorch et al. (2022); Ke et al. (2023) pioneer amortized variational inference for causal discovery. However, these methods rely on high-dimensional embeddings that scale poorly with graph size. SEA (Wu et al., 2025) mitigates this issue by decomposing large graphs into subproblems, but its reliance on classical estimators such as GIES and extensive sub-batch sampling limits inference speed and causes information loss across variable partitions. In contrast, CauScale reduces computation by compressing data embeddings while mitigating information loss via a *data-graph block*, achieving superior efficiency without sacrificing discovery quality.

## 3. Preliminary

**Causal graphical models.** A causal graphical model (CGM) (Peters et al., 2017) consists of (i) a joint distribution $P_X$ over random variables $X = (X_1, \ldots, X_n)$ and (ii) a directed acyclic graph $G = (V, E)$. Each node $i \in V$ corresponds to a variable $x_i$, and each directed edge $(i, j) \in E$ encodes a direct causal influence from $x_i$ to $x_j$. The distribution $P_X$ is *Markov* with respect to $G$, i.e., $p(x_1, \ldots, x_n) = \prod_{j=1}^n p(x_j \mid \text{PA}_j)$, where $\text{PA}_j$ denotes the parent set of node $j$. *Causal sufficiency* is assumed, meaning there are no unobserved common causes that jointly affect multiple variables in $X$.

**Interventions.** CGMs support interventions by modifying the conditional mechanism of target variables. An intervention on node $j$ replaces the conditional distribution

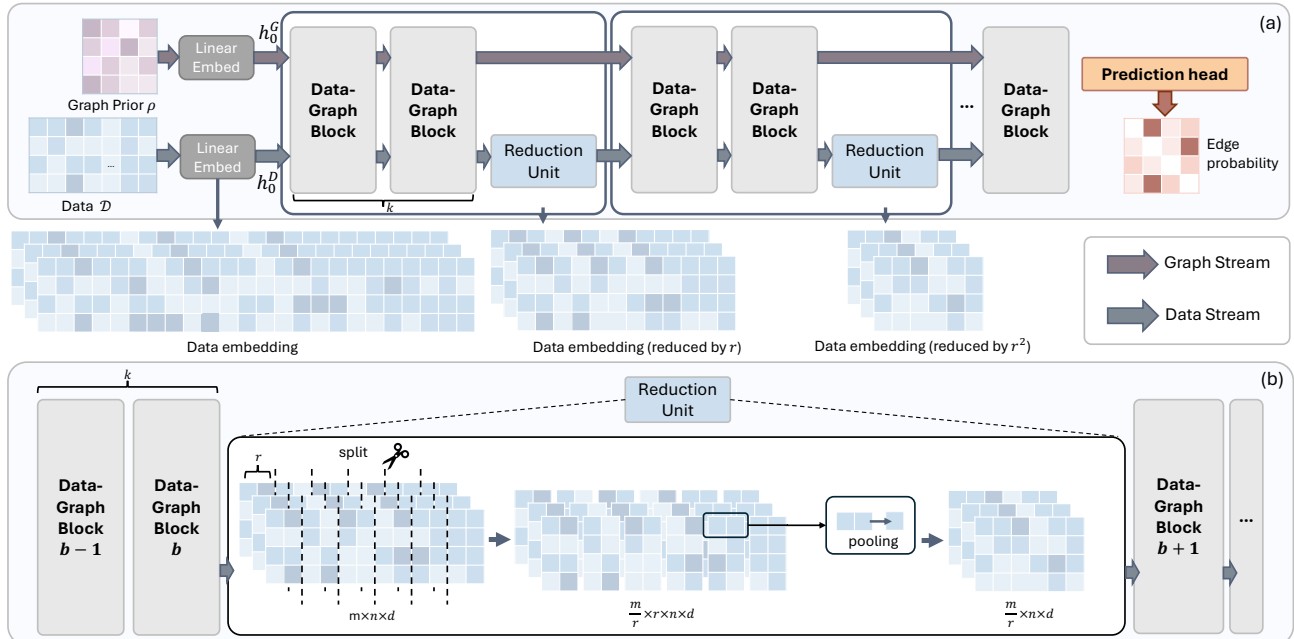

*Figure 1.* The architecture of CauScale. (a) The overall architecture and the changes of data embedding size during network processing. (b) The reduce operation in *reduction unit*. Between each $k$ *data-graph block*s, the *reduction unit* pool the data embedding along the observation dimension to reduce it with a fraction of $r$.

$p(x_j \mid \mathrm{PA}_j)$ with $\tilde{p}(x_j \mid \mathrm{PA}_j)$. Two settings are considered: the *observational* setting (no interventions), and *perfect interventions*, where the intervened variable is randomized independently of its parents, i.e., $\tilde{p}(x_j \mid \mathrm{PA}_j) = \tilde{p}(x_j)$.

## 4. CauScale

### 4.1. Overall Architecture

Let $n$ denote the number of graph nodes and $m$ the number of observational samples. The input data $\mathcal{D} \in \mathbb{R}^{m \times n \times 2}$ concatenates observational variables $D \in \mathbb{R}^{m \times n}$ and a binary intervention indicator $I \in \{0,1\}^{m \times n}$, where $I = 1$ indicates that variable is intervened. The model takes $\mathcal{D}$ and a statistical graph prior $\rho \in \mathbb{R}^{n \times n}$ computed from $D$ as inputs, and outputs a probabilistic adjacency matrix $\hat{G} \in \mathbb{R}^{n \times n}$ representing the likelihood of directed causal relations. The prior $\rho$ is defined as the inverse covariance matrix:

$$\rho = \left( \mathbb{E}\big[(D - \mu)(D - \mu)^\top\big] \right)^{-1}, \mu = \mathbb{E}[D]$$

The inputs $\mathcal{D}$ and $\rho$ are encoded into initial embeddings $h^{\mathcal{D}} \in \mathbb{R}^{m \times n \times d}$ and $h^G \in \mathbb{R}^{n \times n \times d}$ via linear layers, where $d$ is the embedding dimension. These embeddings are then processed by alternating stacks of *data-graph block* and *reduction unit*. Each *data-graph block* updates both the data and graph streams (Section 4.2). Every $k$ blocks, the *reduction unit* pools the data-stream embedding along the sample dimension to reduce its length by a factor of $r$ (Sec-

tion 4.3). Within each *data-graph block*, we employ tied attention weights to reduce memory overhead (Section 4.4). Finally, the graph-stream output is fed into a prediction head to produce $\hat{G}$ (Section 4.5). Figure 1 shows the overall architecture of CauScale.

### 4.2. DataGraph Block

As shown in Figure 2, each *data-graph block* consists of three modules: a data layer, a data2graph layer, and a graph layer. Given the incoming data and graph embeddings $(h_{b-1}^D, h_{b-1}^G)$, the block proceeds in three steps: (1) The data layer updates the data stream embedding, producing $h_b^D$. This updated embedding is forwarded to the next *data-graph block* and, when applicable, to the *reduction unit* for compression. (2) The data2graph layer summarizes $h_b^D \in \mathbb{R}^{m \times n \times d}$ into an observation-compressed relation matrix $\omega_b^{D \to G} \in \mathbb{R}^{n \times n}$, which captures node relationship information. (3) The graph layer injects this message into the graph stream by concatenating $\omega_b^{D \to G}$ with the previous graph embedding $h_{b-1}^G \in \mathbb{R}^{n \times n \times d}$, and produces the updated graph embedding $h_b^G$.

**Data2Graph layer.** The data2graph layer extracts pairwise relational evidence from the data stream and summarizes it into a graph message. Given the data embedding $h_b^D \in \mathbb{R}^{m \times n \times d}$, we first apply a data axial-attention layer to obtain $h^{D \to G} \in \mathbb{R}^{m \times n \times d}$. We then map $h^{D \to G}$ to two node-level embeddings $u^{D \to G}, v^{D \to G} \in \mathbb{R}^{n \times d}$ using two

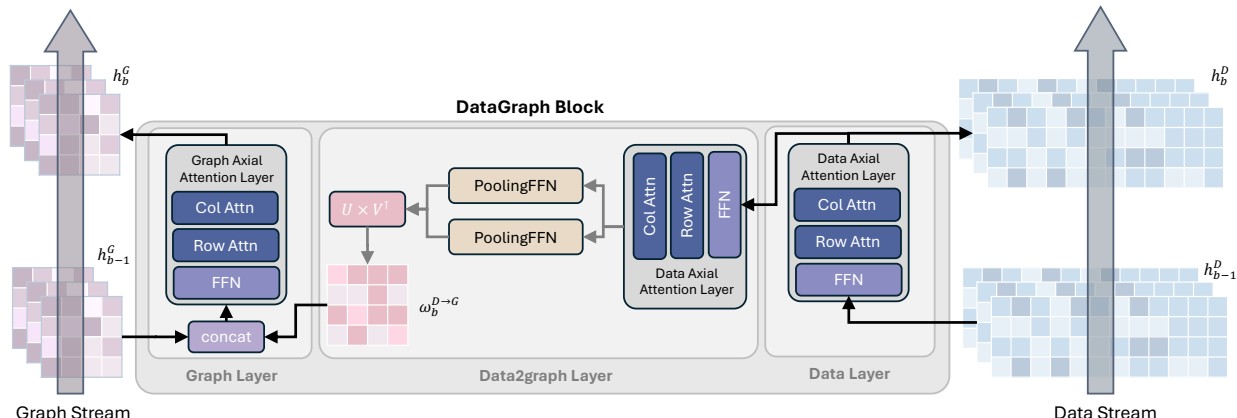

*Figure 2.* Structure of the *DaraGraph Block*. The *data-graph block* process information on data and graph stream. On data stream, after being processed by the data axial attention layer, data embedding $h_b^D$ is sent to both the next module on data stream and summarized by the data2graph layer to graph message $\omega_b^{D \to G}$. The message will be concatenated with previous graph embedding $h_{b-1}^G$ and processed by graph layer in graph stream.

separate PoolingFFN modules, each performing average pooling over the observation dimension followed by an MLP. Finally, we form $\omega_b^{D \to G} = u^{D \to G} (v^{D \to G})^\top \in \mathbb{R}^{n \times n}$, which represents directed pairwise relations between variables.

**Graph layer.** The graph layer injects $\omega_b^{D \to G}$ into the graph stream by concatenating it with the previous graph embedding $h_{b-1}^G \in \mathbb{R}^{n \times n \times d}$, yielding $h_b^{G'} \in \mathbb{R}^{n \times n \times (d+1)}$. A linear projection maps $h_b^{G'}$ back to $\mathbb{R}^{n \times n \times d}$, which is then processed by a graph axial-attention layer to produce the updated graph embedding $h_b^G \in \mathbb{R}^{n \times n \times d}$.

## 4.3. Reduction Unit

Naively subsampling observations for estimation can discard informative samples and degrade causal discovery. Instead, we compress the *data-stream embedding* during network processing, reducing computation while preserving the variable-wise embedding. This design is motivated by three considerations. (1) **Efficiency:** in typical causal discovery datasets, the number of observational samples $m$ is often one to three orders of magnitude larger than the number of nodes $n$. Compressing along the observation dimension therefore yields substantial computational savings. (2) **Dependency structure:** causal signals are primarily expressed through dependencies among nodes within each observational sample. Under the standard i.i.d. assumption across samples, aggregating embeddings across observational samples is generally less destructive than collapsing the variable dimension. (3) **Reduced information loss:** the reduction is applied after several *data-graph block*s have transformed raw inputs into more informative representations. Moreover, a Data2Graph module is executed before reduction to distill local relational signals into the graph stream, allowing the data stream to be compressed without losing critical

structural evidence.

Accordingly, CauScale applies the *reduction unit* every $k$ *data-graph block*s[1]. Given a data embedding $h_b^D \in \mathbb{R}^{m \times n \times d}$ after block $b \in \{0, \dots, B-1\}$ and a reduction factor $r$, we group the observation dimension into chunks of size $r$ and average-pool within each chunk. When $r \nmid m$, we set $\hat{m} = r \lfloor m/r \rfloor$ and discard the last $m - \hat{m}$ samples for convenience (replacing $m$ with $\hat{m}$). Specifically, we reshape

$$h_b^D : \ m \times n \times d \ \to \ \tfrac{m}{r} \times r \times n \times d,$$

and apply average pooling over the group dimension of size $r$, yielding the reduced embedding $\tilde{h}_b^D \in \mathbb{R}^{\frac{m}{r} \times n \times d}$.

## 4.4. Tied Attention Weights

The core component in each stream within a *data-graph block* is an axial-attention layer, which applies self-attention along two axis (row-wise and column-wise), followed by an FFN. Each sub-layer is wrapped with layer normalization, dropout, and residual connections. To improve space efficiency, we adopt the tied attention weight mechanism from Rao et al. (2021), which avoids maintaining axis-specific attention maps and substantially reduces attention memory. For illustration, consider attention along row-axis with $Q, K, V \in \mathbb{R}^{R \times C \times H \times d_{\text{head}}}$, where $R$ and $C$ denote the row and column dimensions (e.g., $R{=}m$, $C{=}n$ for the data stream), $H$ is the number of heads, and $d_{\text{head}}$ is the head dimension. Following Rao et al. (2021), we tie attention weights across rows and only store $A \in \mathbb{R}^{H \times C \times C}$, while

---

[1] In practice, the first reduction is applied after $k+1$ *data-graph block*, allowing the network to build more expressive representations before compression begins.

keeping the output shape unchanged:

$$A_{h,i,j} = \sum_{r=1}^{R} \sum_{t=1}^{d_{\text{head}}} Q_{r,i,h,t} \cdot K_{r,j,h,t},$$

$$O_{r,i} = W^O \cdot \left[ \sum_{j=1}^{C} \text{softmax}_j(A_{h,i,j}) \cdot V_{r,j,h,:} \right]_h + b^O,$$

### 4.5. Prediction Head

After the final *data-graph block*, we take the graph-stream output $h_{B-1}^G \in \mathbb{R}^{n \times n \times d}$, apply layer normalization, and feed it into a pairwise graph prediction head. Following Wu et al. (2025) and Lippe et al. (2021), we do not explicitly enforce acyclicity during prediction, since imposing DAG constraints typically requires additional constrained optimization or post-processing and can be computationally expensive. Moreover, real-world data sometimes contain cycles. We adopt the decomposed head in Wu et al. (2025). For each unordered node pair $\{i, j\}$ with $i < j$, we compute logits over three edge states (no edge, $i \to j$, $j \to i$) by

$$g_{\{i,j\}} = \text{FFN}\big([h_{B-1,i,j}^G, h_{B-1,j,i}^G]\big) \in \mathbb{R}^3, \quad (1)$$

where $[\cdot, \cdot]$ denotes concatenation. Collecting all pairs yields $g \in \mathbb{R}^{\frac{N(N-1)}{2} \times 3}$, and we obtain probabilities via a softmax over the three states for each pair. In our experiments, this decomposed head achieves accuracy comparable to the AVICI prediction head while empirically producing fewer cycles in the decoded graphs.

### 4.6. Efficiency Analysis

**Time efficiency.** The dominant cost of data-stream axial attention comes from two terms: (i) *sample-axis* attention over $m$ samples for each of the $n$ variables, with cost $\mathcal{O}(nm^2)$; and (ii) *node-axis* attention over $n$ variables for each of the $m$ samples, with cost $\mathcal{O}(mn^2)$. With a reduction factor $r$ applied every $k$ blocks, the effective sample length at block $b$ becomes $m_b = m/r^{\lfloor b/k \rfloor}$. The average per-block compute is therefore

$$\mathcal{C}_{\text{sample}} \propto \frac{1}{B} \sum_{b=0}^{B-1} nm_b^2 = \frac{nm^2}{B} \sum_{b=0}^{B-1} r^{-2\lfloor b/k \rfloor},$$

$$\mathcal{C}_{\text{node}} \propto \frac{1}{B} \sum_{b=0}^{B-1} n^2 m_b = \frac{n^2 m}{B} \sum_{b=0}^{B-1} r^{-\lfloor b/k \rfloor}.$$

When $B$ is a multiple of $k$, these sums reduce to geometric series: $\sum_{b=0}^{B-1} r^{-2\lfloor b/k \rfloor} = k \sum_{i=0}^{B/k-1} r^{-2i}$ and $\sum_{b=0}^{B-1} r^{-\lfloor b/k \rfloor} = k \sum_{i=0}^{B/k-1} r^{-i}$. In our experiments $(B=10, k=2, r=2)$, given that the first reduction is delayed by one block, this yields 36.60% of the baseline sample-axis compute and 48.13% of the baseline node-axis compute.

**Space efficiency.** Given attention on row axis, standard attention mechanism stores axis-specific attention maps $A \in \mathbb{R}^{R \times H \times C \times C}$, resulting in $\mathcal{O}(RHC^2)$ memory. With tied attention weights (Rao et al., 2021), attention weights are shared across target axis and only $A \in \mathbb{R}^{H \times C \times C}$ is stored, reducing attention-map memory to $\mathcal{O}(HC^2)$. Analogously, for column-axis attention, the memory cost is reduced from $\mathcal{O}(CHR^2)$ to $\mathcal{O}(HR^2)$.

## 5. Experiment

### 5.1. Settings

**Baselines.** We evaluate our approach against several baselines spanning different paradigms: (1) constraint-based methods: Fast Causal Inference (FCI) (Spirtes et al., 2013); (2) score-based methods: NOTEARS (Zheng et al., 2018), SDCD (Nazaret et al.) (3) FCM-based methods: DiffAN (Sanchez et al., 2023) (4) pre-training-based methods: AVICI (Lorch et al., 2022), SEA (Wu et al., 2025). We additionally include two fundamental statistical measures as reference points: global Pearson correlation (CORR) (Benesty et al., 2009) and inverse covariance matrix (INVCOV) (Hartlap et al., 2007).

**Evaluation metrics.** To evaluate causal discovery performance, we adopt eight metrics: Structural Hamming Distance (SHD), Mean Average Precision (mAP), Area Under the ROC Curve (AUC), Orientation Accuracy (OA), Cyclicity, and the AID metric family (Ancestor-AID, O-set-AID, Parent-AID) (Henckel et al., 2024). To evaluate causal discovery efficiency, we report two metrics: inference time and peak GPU memory.

### 5.2. Datasets

We consider two types of data: synthetic datasets generated from SCMs and semi-synthetic single-cell expression datasets simulated from gene regulatory networks (GRNs).

**Training data.** For synthetic data, we generate datasets based on Erdős-Rényi and Scale-Free graphs. The graph size $n$ ranges from 10 to 500, with edge counts $|E| \in \{n, 2n, 3n, 4n\}$. The causal mechanisms include both linear and neural network (NN) functions with additive or non-additive Gaussian noise. For each graph, we sample 1,000 observations, consisting of observational and single-node interventional data in a $1 : n$ ratio. For single-cell GRNs, we utilize the SERGIO GRN simulator (Dibaeinia & Sinha, 2020) to generate gene expression data. The underlying graph topologies are initialized using Erdős-Rényi, Scale-Free, and Stochastic Block Models. Given the complexity of gene regulatory dynamics, we increase the sample size to 5,000 to ensure reliable structure learning. Consequently, to balance the computational overhead introduced by this larger sample size, we restrict the maximum graph size to $n = 200$. Details are provided in Appendix B.

**Tesing data.** We construct separate benchmarks to assess scalability and robustness. For synthetic data, we assess

*Table 1.* Model performance comparison. † indicates o.o.d settings. Time represents inference time. Note: DiffAN and FCI are excluded from large node or large sample size settings due to excessive time costs. '-' indicates that OA is not applicable, as CORR and INVCOV output symmetric matrices and thus provide no edge orientation.

**Synthetic** (sample size = 1000)

| Model | Linear | | | | NN non-add. | | | | Sigmoid† | | | | Polynomial† | | | | Time |
|---|---|---|---|---|---|---|---|---|---|---|---|---|---|---|---|---|---|
| | mAP | SHD | AUC | OA | mAP | SHD | AUC | OA | mAP | SHD | AUC | OA | mAP | SHD | AUC | OA | (s) |
| *Setting: n = 100, \|E\| = 400* | | | | | | | | | | | | | | | | | |
| CORR | 20.1 | 578.0 | 79.8 | - | 14.7 | 605.4 | 74.0 | - | 28.4 | 501.2 | 86.7 | - | 23.1 | 532.8 | 76.9 | - | 0.0008 |
| INVCOV | 34.1 | 491.2 | 93.6 | - | 23.6 | 530.8 | 82.5 | - | 32.9 | 477.8 | 90.5 | - | 23.7 | 504.6 | 72.7 | - | 0.0275 |
| FCI | 12.4 | 372.0 | 55.3 | 10.7 | 11.1 | 359.8 | 55.3 | 11.0 | 15.1 | 348.2 | 56.3 | 12.6 | 9.0 | 368.4 | 53.0 | 6.1 | 84.987 |
| NOTEARS | 29.4 | 300.8 | 51.8 | 27.8 | 17.8 | 337.0 | 50.4 | 19.5 | 11.3 | 366.0 | 49.1 | 8.3 | 8.6 | 371.8 | 52.5 | 4.9 | 2170.2 |
| SDCD | 42.3 | 400.2 | 89.3 | 81.6 | 65.7 | 272.6 | 89.0 | 79.8 | 61.9 | 327.8 | 87.6 | 76.5 | 41.9 | 303.8 | 70.9 | 42.8 | 67.428 |
| DiffAN | 10.6 | 475.4 | 51.3 | 8.5 | 8.4 | 465.2 | 53.6 | 9.3 | 12.3 | 389.4 | 57.4 | 12.3 | 11.6 | 378.9 | 50.3 | 11.1 | 1973.4 |
| AVICI | 25.9 | 394.0 | 80.2 | 81.4 | 32.3 | 361.2 | 81.1 | 81.5 | 22.7 | 371.8 | 68.4 | 63.2 | 5.6 | 384.6 | 45.9 | 36.8 | 0.2974 |
| SEA-gies | 92.1 | 108.6 | 99.2 | 94.8 | 51.2 | 306.2 | 88.4 | 86.6 | 72.7 | 192.2 | 92.2 | 85.4 | 36.2 | 319.2 | 74.2 | 70.8 | 8.7759 |
| CauScale (Ours) | **99.6** | **15.2** | **100.0** | **100.0** | **89.0** | **105.6** | **98.5** | **99.5** | **84.4** | **125.8** | **95.0** | **94.6** | **50.3** | **252.2** | **79.4** | **81.7** | **0.0384** |
| *Setting: n = 1000, \|E\| = 2000†* | | | | | | | | | | | | | | | | | |
| CORR | 34.3 | 2376.6 | 99.5 | - | 16.2 | 3031.0 | 93.9 | - | 34.7 | 2304.2 | 97.7 | - | 25.0 | 2472.6 | 83.7 | - | 0.0455 |
| INVCOV | 46.7 | 1996.6 | 99.8 | - | 28.0 | 2432.2 | 92.6 | - | 40.6 | 2056.6 | 97.9 | - | 26.4 | 2359.0 | 83.9 | - | 0.4412 |
| FCI | 32.9 | 1309.0 | 67.2 | 34.4 | 12.7 | 1721.2 | 58.7 | 17.4 | 8.6 | 1828.6 | 54.5 | 9.0 | 1.5 | 2008.8 | 50.7 | 1.4 | 2005.2 |
| NOTEARS | 30.5 | 1388.6 | 50.6 | 30.5 | 20.5 | 1677.8 | 50.0 | 23.2 | 11.0 | 1790.4 | 50.0 | 10.9 | 7.3 | 1893.2 | 53.6 | 7.1 | 10896 |
| SDCD | 54.1 | 1793.2 | 99.3 | **98.6** | 59.6 | 2015.0 | 87.9 | 76.0 | 48.5 | 1649.2 | 89.2 | 78.5 | **29.8** | **1798.4** | 74.6 | 49.2 | 65.386 |
| AVICI | 0.2 | 1985.8 | 47.5 | 46.2 | 0.9 | 1980.2 | 56.8 | 54.6 | 0.2 | 2006.8 | 39.6 | 41.9 | 0.1 | 2037.0 | 37.0 | 40.3 | 3.3407 |
| SEA-gies | 66.3 | 2944.8 | 98.2 | 80.2 | 11.9 | 3227.4 | 73.9 | 66.2 | 48.1 | 1359.0 | 88.8 | 70.1 | 20.6 | 6814.2 | 72.0 | 58.9 | 218.23 |
| CauScale (Ours) | **96.6** | **230.0** | **100.0** | 96.5 | **79.7** | **835.0** | **98.2** | **96.6** | **64.5** | **1064.6** | **95.3** | **79.0** | 18.9 | 3985.0 | **78.1** | **59.7** | 0.8288 |

**SERGIO-GRN** (sample size = 20000)

| Model | n = 100, \|E\| = 400 | | | | | n = 200, \|E\| = 400 | | | | |
|---|---|---|---|---|---|---|---|---|---|---|
| | mAP | SHD | AUC | OA | Time | mAP | SHD | AUC | OA | Time |
| CORR | 4.5 | 403.4 | 51.9 | - | 0.0128 | 1.1 | 409.8 | 52.7 | - | 0.0147 |
| INVCOV | 5.1 | 403.0 | 52.3 | - | 0.0793 | 1.2 | 409.8 | 52.4 | - | 0.0995 |
| NOTEARS | 4.1 | 492.8 | 50.5 | 2.0 | 5040.8 | 1.0 | 591.0 | 49.8 | 0.4 | 5805.9 |
| SDCD | 4.0 | 764.6 | 49.0 | 5.1 | 579.68 | 2.1 | 1265.0 | 49.9 | 3.7 | 670.10 |
| AVICI | Out of memory | | | | | Out of memory | | | | |
| SEA-gies | 5.6 | 400.2 | 58.8 | 61.6 | 11.769 | 1.2 | 407.8 | 55.0 | 57.07 | 17.288 |
| CauScale (Ours) | **54.0** | **290.8** | **90.5** | **94.2** | **1.3058** | **39.2** | **336.8** | **90.8** | **90.9** | **2.5195** |

the model on graphs of varying scales, with $(n, |E|) \in \{(100, 400), (1000, 2000)\}$ and a sample size of 1,000. We also introduce two out-of-distribution causal mechanisms: sigmoid and polynomial functions. For GRN data, we evaluate on graphs with $(n, |E|) \in \{(100, 400), (200, 400)\}$, using a larger sample size of 20,000. For each testing configuration, we generate 5 independent Erdős-Rényi graph instances, and report the averaged results.

### 5.3. Model Performance

Table 1 compares CauScale against other baselines on mAP, SHD, AUC and OA, with results averaged over 5 test graphs per setting. CauScale demonstrates superior accuracy and efficiency across varying graph sizes and mechanisms. AID results are reported in Appendix F.1.

**Accuracy.** On the synthetic dataset with $n = 100$, our model achieves near-perfect causal discovery on linear data (99.6% mAP) and consistently outperforms baselines in non-linear settings. Notably, on large-scale graphs with 1000 nodes (a size unseen during training), CauScale maintains high performance (96.6% mAP for linear), despite our model being trained on graphs with at most 500 nodes. CauScale also exhibits strong generalization capabilities on OOD mechanisms. Specifically, on the polynomial dataset ($n = 100$) with more complex mechanisms, CauScale achieves 50.3% mAP, whereas the second and third best methods, SEA and SDCD, drop to 36.2% and 41.9%, respectively. On the SERGIO-GRN dataset, CauScale achieves the best performance across all metrics and settings among all causal discovery baselines.

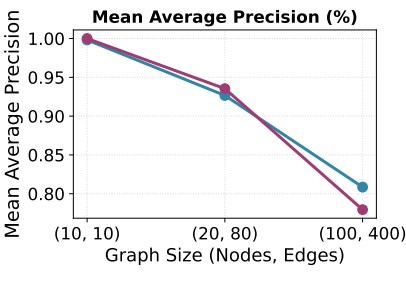
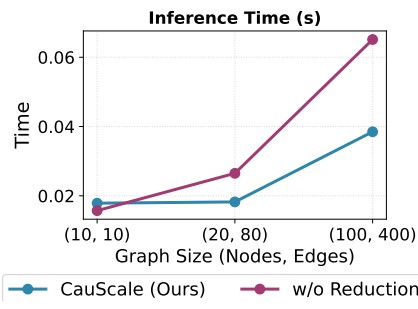
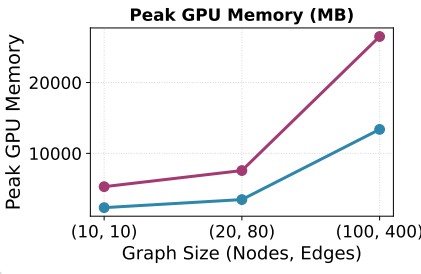

*Figure 3.* Comparison of w/ and w/o Reduction Unit.

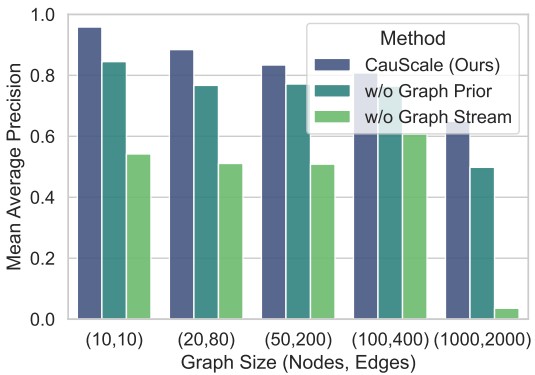

*Figure 4.* Advantage of *data-graph block* over the block containing the data layer only.

*Table 2.* Cyclicity and cycle-breaking impact for `CauScale` at $n = 1000, |E| = 2000$ (o.o.d. graph size). Cyclicity denotes the fraction of predicted graphs containing at least one cycle. All in-distribution graph-size settings ($n \leq 500$) and SERGIO-GRN produce zero cycles.

|  | Linear | NN | Sigmoid$^\dagger$ | Poly.$^\dagger$ |
|---|---|---|---|---|
| Cyclicity | 0.2 | 0.4 | 0.6 | 1.0 |
| Edges removed (avg.) | 0.4 | 0.6 | 0.6 | 103 |
| SHD (original) | 230.0 | 835.0 | 1064.6 | 3985.0 |
| SHD (after cycle-break) | 230.0 | 834.6 | 1064.4 | 3891.8 |

**Time and space efficiency.** `CauScale` achieves the shortest inference time among all evaluated causal discovery algorithms. Even on graphs with $n = 1000$, our inference takes less than 1 second (0.8288s), achieving a speedup of over $13,000\times$ compared to NOTEARS (10,896s), $200\times$ compared to SEA-gies (218s), and $4\times$ compared to AVICI (3.34s). Regarding space efficiency, on SERGIO-GRN dataset, AVICI fails with an Out-of-Memory error even at $n = 100$. In contrast, `CauScale` successfully scales to $n = 200$ with 20,000 samples.

**Cyclicity.** `CauScale` produces zero cycles across all in-distribution graph-size settings (synthetic $n \leq 500$ and SERGIO-GRN). At $n = 1000$—twice the largest training graph size—cycles emerge. To address cycles in such hard cases, we implement a cycle-breaking strategy that iteratively removes the edge with the lowest predicted score in each detected cycle until the graph is acyclic. We report SHD and the average number of edges removed per graph before and after applying the procedure in Table 2. For the polynomial case, an average of 103 edges per graph are removed by cycle-breaking. This is expected given that both graph size and causal mechanism fall outside the training distribution. For all other settings, cycle-breaking removes at most 0.6 edges per graph on average. Applying this cycle-breaking procedure consistently improves or

preserves SHD.

### 5.4. Ablation Studies

**W/ and w/o reduction unit** We remove the *reduction unit* and retrain the model on synthetic dataset to validate its importance. Since the network encounters Out-of-Memory errors on the original training set without the *reduction unit*, we use a training subset with node number limited to $n \in \{10, 20, 100\}$. We train both `CauScale` and the architecture w/o *reduction unit* on this subset and evaluate their performance on synthetic test set with the same node number. Other settings in test set are the same with the evaluation benchmark. Results are averaged across all four distributions (linear, NN, sigmoid, and polynomial). Figure 3 illustrates the mean average precision, inference time, and peak GPU memory usage. The benefits of the *reduction unit* become increasingly pronounced as the number of nodes increases, enabling the model to maintain high accuracy while achieving significantly faster inference speeds and lower GPU memory usage. Note that at $n = 100$, using *reduction unit* is considerably better than without it. We attribute this to the training efficiency brought by *reduction unit*. Without it, the network must process all $m$ samples throughout every layer, leading to higher training complexity and potentially degraded optimization. The reduction unit compresses redundant or noisy signals into more compact and informative representations, progressively reducing the computational burden and allowing the network to learn more effective representations.

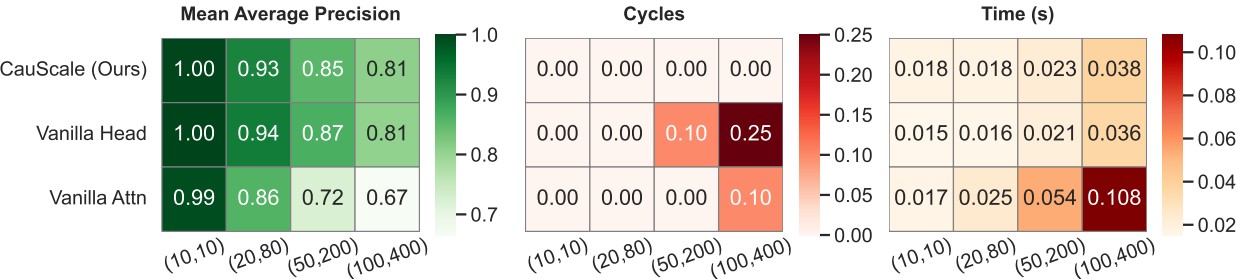

*Figure 5.* Ablation on components: Ours vs. AVICI.

*Table 3.* Mean Average Precision (%) comparison across different pooling strategies in the Reduction Unit. We use average pooling in `CauScale`.

| (Nodes, Edges) | Max | Strided | Average |
|---|---|---|---|
| (10,10) | 94.3 | 100.0 | 100.0 |
| (20,80) | 76.0 | 84.6 | 92.6 |
| (50,200) | 71.2 | 78.5 | 85.4 |

**Graph components** We conduct ablation studies by (1) removing the input graph prior by setting the graph input to an all-ones vector (w/o Graph Prior) and (2) removing the graph stream while retaining only the data stream (w/o Graph Stream). We retrain the two ablation versions on our synthetic train set. Figure 4 demonstrates the performance comparison on our synthetic benchmark. Removing the graph stream causes the most significant performance degradation, highlighting the importance of it. Removing the graph prior causes less performance degradation but still yields inferior results compared to our full model, validating the importance of the inductive bias.

**Attention shape and output head** We conduct an ablation study to analyze the components of our model relative to AVICI. Specifically, we evaluate two variants: (1) replacing the tied-attention mechanism in `CauScale` with the vanilla attention from (Lorch et al., 2022) (Vanilla Attn), and (2) replacing our prediction head in Equation 1 with the vanilla prediction head from (Lorch et al., 2022) (Vanilla Head). The latter projects the graph embedding $h_{B-1}^G \in \mathbb{R}^{n \times n \times d}$ to a logit $g \in \mathbb{R}^{n \times n}$ using a Feed-Forward Network, followed by a sigmoid function to obtain edge probabilities. Due to the high space complexity of vanilla self-attention, we limited this study to a subset of the training set with node counts $n \in \{10, 20, 100\}$. Figure 5 illustrates the results averaged across all causal mechanisms. The results show the advantage of our implemented components. First, the tied-attention mechanism demonstrates superior computational efficiency, achieving an inference speed six times faster than

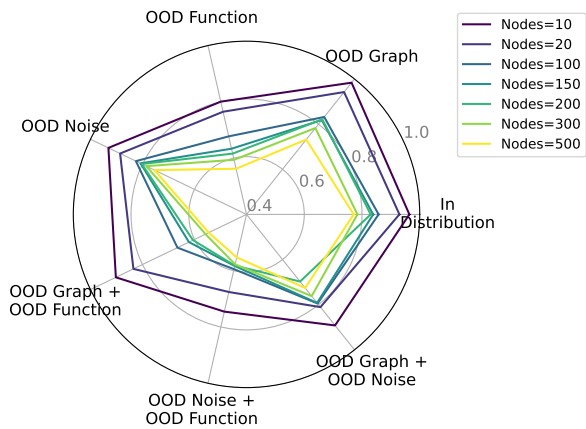

*Figure 6.* Generalization property of `CauScale` on OOD graphs, noise, and mechanism functions.

vanilla attention on graphs with $(n = 100, |E| = 400)$. It also yields higher mean average precision score, striking a perfect balance between efficiency and accuracy. Second, our pairwise processing head leads to a much more lower degree of cyclicity (0.0%) compared to the vanilla prediction head used in AVICI (0.0-0.25%).

**Pooling strategy of reduction unit** We conduct a lightweight ablation study using graphs with node counts of $n = \{10, 20, 50\}$ during training and testing to efficiently evaluate different pooling strategies within the *reduction unit*. In addition to the average pooling employed in `CauScale`, we compare two alternative downsampling techniques: strided pooling and max pooling. Table 3 demonstrates that average pooling consistently achieves superior performance across experimented graph sizes.

### 5.5. Generalization Analysis

We further evaluate the generalization capability of the synthetic-data-trained model described in Section 5.3 across OOD graph structures generated by Stochastic Block Mod-

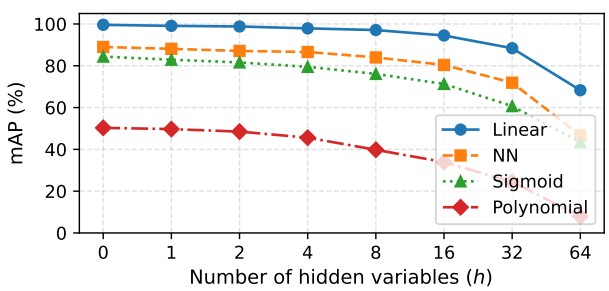

*Figure 7.* Model performance (mAP %) under varying numbers of hidden variables $h$, with $n_{obs} = 100 - h$ observed variables ($n = 100$, $|E| = 400$).

els, OOD noise distributions (uniform and Laplace), and OOD functions (sigmoid and polynomial). Figure 6 reveals that the model demonstrates strong generalization to OOD graph structures but exhibits greater sensitivity to OOD noise patterns and mechanism functions. These findings indicate that future model training should prioritize generating datasets with more diverse noise distributions and mechanism functions to enhance robustness.

### 5.6. Robustness to Latent Confounders

We test the model robustness under latent confounder. We simulate latent confounding via selective node removal. Specifically, we hide the top-$h$ nodes ranked by out-degree in graphs with $n = 100$, as high out-degree nodes act as strong common causes. After removal, the observable sub-graph contains $n_{obs} = 100 - h$ nodes. We vary $h$ as 1, 2, 4, 8, 16, 32, 64 to cover a wide range of confounding severity. Table 7 reports mAP (%) on the observable sub-DAG. Under mild confounding ($h \leq 8$), the model remains robust across all mechanisms, with linear mAP staying above 97%. Under severe confounding ($h = 64$), performance drops substantially, which is expected given that the observable sub-graph loses much of its causal structure.

### 5.7. Robustness to Graph Prior Error

We evaluate model sensitivity to the graph prior by conducting a controlled experiment that varies the number of samples used to compute the prior (from 10 to 500), using the same model and settings as the main experiment, across graph sizes ranging from $n = 10$ to $n = 1000$. Results are summarized in Table 4. Two patterns emerge: (1) the model remains robust to a noisy prior on smaller graphs; (2) sensitivity to prior quality increases with graph size.

### 5.8. Sample Size analysis

We conduct a sample size analysis on our trained models in Section 5.3, with inference sample sizes varying from 500 to 4000 for synthetic data and from 1000 to 20000 for

*Table 4.* Model performance (mAP %) under varying numbers of samples used to compute the graph prior.

| | Number of Samples for Prior | | | |
| Graph Size | 10 | 100 | 200 | 500 |
|---|---|---|---|---|
| $n = 10$, $|E| = 10$ | 85.9 | 94.2 | 95.7 | 95.8 |
| $n = 100$, $|E| = 400$ | 75.6 | 77.9 | 79.0 | 80.8 |
| $n = 1000$, $|E| = 2000$ | 32.6 | 48.1 | 58.1 | 64.9 |

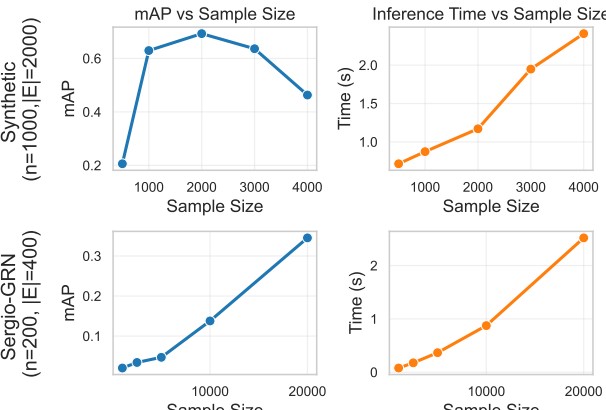

*Figure 8.* Sample size analysis by mean average precision.

SERGIO-GRN data. Figure 8 shows that the model trained on synthetic data achieves peak performance with sample size of 2000, while the model for SERGIO-GRN achieves the best performance with a sample size of 20000. This suggests that more complex causal mechanisms necessitate larger sample sizes for accurate inference.

## 6. Conclusion

We presented CauScale, an efficient neural architecture for large-scale causal discovery that addresses the time and memory bottlenecks of prior methods. CauScale combines a reduction unit for time efficiency, tied attention weights for space efficiency, and a two-stream design that preserves structural signals under compression. Extensive experiments on synthetic and semi-synthetic single-cell benchmarks show that CauScale scales training to 500-node graphs and enables inference on graphs with up to 1,000 nodes, achieving strong accuracy with $4\times$–$13,000\times$ speedups over existing approaches. These results suggest a practical direction for pre-training efficient neural models for causal discovery at scale. While CauScale demonstrates strong performance on synthetic and semi-synthetic benchmarks, evaluation on real-world observational datasets remains an important direction for future work. The reliance on simulated training data may limit generalization to settings where the data-generating process deviates significantly from the assumed SCM framework.

## Acknowledgements

This work is supported in part by the Shanghai Artificial Intelligence Laboratory and in part by the National Natural Science Foundation of China under Grant 625B2131.

## Impact Statement

This paper presents `CauScale`, a neural architecture for efficient and scalable causal discovery on large graphs. The primary impact is to make causal structure learning more computationally accessible—reducing runtime and memory requirements—and thereby supporting faster hypothesis generation in data-intensive scientific domains. As with any causal discovery method, results can be sensitive to data quality, modeling assumptions, and distribution shift; spurious edges may arise if the input data are noisy or misspecified. We emphasize that the predicted graphs should be treated as hypotheses and validated by domain experts and, where applicable, downstream experiments before being used in high-stakes decision-making.

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

## A. Evaluation metrics.

To evaluate causal discovery performance, we adopt the following metrics for causal structure learning: (1) **Structural Hamming Distance (SHD)**: the minimum number of edge insertions, deletions, and reversals required to transform the predicted graph into the ground-truth graph. (2) **Mean Average Precision (mAP)**: the area under the precision-recall curve computed over all candidate edges, averaged across the graph. (3) **Area Under the ROC Curve (AUC)**: the area under the ROC curve computed over all candidate edges, averaged across the graph. (4) **Orientation Accuracy (OA)**: the fraction of ground-truth directed edges for which the model assigns a higher probability to the correct direction. (5) **Cyclicity**: the fraction of predicted graphs containing at least one directed cycle. (6) **Adjustment Identification Distance (AID)** (Henckel et al., 2024): a family of three metrics (Ancestor-AID, O-set-AID, Parent-AID) that measure the fraction of causal effects that would be incorrectly inferred from the predicted graph, each corresponding to a different adjustment strategy.

## B. Data Generation Details

### B.1. Causal Graph Details

We evaluate our method on various causal graphs. Below, we provide a detailed description of the graph models used in this study. Both the synthetic data and SERGIO-GRN data are generated based on these Directed Acyclic Graph (DAG) structures.

- **Erdős-Rényi (ER):** A standard random graph model where edges are added between any pair of nodes with a fixed probability $p$. This results in a graph where the degree distribution is approximately Poissonian, representing networks with uniform connectivity patterns.

- **Scale-Free (SF):** Generated using the Barabási-Albert preferential attachment process. New nodes are more likely to attach to existing nodes with high degrees. This topology creates networks with "hubs" and follows a power-law degree distribution, simulating real-world biological (e.g., gene regulatory networks) or social networks.

- **Stochastic Block Model (SBM):** A generative model for graphs with community structure. Nodes are assigned to one of $K$ latent blocks (clusters). Edge probabilities depend on the block membership of the nodes (high probability within blocks, low probability between blocks). This is particularly useful for modeling modular systems, such as protein-protein interaction networks with functional modules.

### B.2. Synthetic Data Generation

The details of the distribution settings for the synthetic data are explained below. We generate data following the code and settings in Wu et al. (2025) and Brouillard et al. (2020). Let $X_i$ denote the target node, $\text{PA}_i$ its parents, $N_i$ an independent noise variable, and $W$ the randomly initialized weights.

- **Linear**: The most fundamental assumption where dependencies are linear: $X_i = W_i\text{PA}_i + N_i$.

- **Neural Networks (NN-Add)**: The mechanism follows $X_i = \text{MLP}(\text{PA}_i) + N_i$, where MLP is a random initialized Multi-Layer Perceptron (MLP) with a single hidden layer and nonlinear activations (PReLU).

- **Neural Networks (NN)**: The noise is concatenated with the parents as input to the neural network: $X_i = \text{MLP}(\text{PA}_i, N_i)$.

- **Sigmoid Additive**: $X_i = \sum W_j \sigma(PA_{ij}) + N_i$, simulating biological saturation effects.

- **Polynomial**: $X_i = \sum_{k=0}^{2} W_k \text{PA}_i^k + N_i$, modeling polynomial dependencies.

Root nodes are initialized using a Uniform distribution of Uniform(-1,1). We set noise to $N_i \sim 0.4 \cdot \mathcal{N}(0, \sigma^2)$, where $\sigma^2 \sim \text{Uniform}(1, 2)$. We apply interventions (hard intervention) one node at a time, covering all node and setting their mechanisms to Uniform(-1,1). For smaller graphs ($N \in \{10, 20, 100\}$), we generate 600 distinct graph structures for each parameter combination. For larger graphs ($N \in \{150, 200, 300, 500\}$), we generate 300 distinct structures per combination due to the increasing computational time required for generation.

### B.3. SERGIO-GRN Data Generation

We generated the data using a slightly modified version of the SERGIO-GRN (Dibaeinia & Sinha, 2020) code from AVICI (Lorch et al., 2022). The simulator generates gene expression data by sampling from the steady state of a dynamic system, described by Stochastic Differential Equations (SDEs) (Dibaeinia & Sinha, 2020). Downstream regulatory interactions are modeled using Hill functions (Chu et al., 2009), ensuring the realistic gene behavior. All samples are generated under gene intervention settings. Specifically, we conduct gene knockouts by setting the target gene expression level to zero. Regarding graph structures, we employ Erdős-R'enyi (ER), Scale-Free, and Stochastic Block Models for training. The number of cell types is set between 5 and 10. We generate 200 distinct graph structures for each setting. We consider training graphs with sizes $N \in \{10, 20, 30, 50, 80, 100, 150, 200\}$, where the number of edges $E \in \{2N, 4N, 6N\}$.

## C. Baseline Implementation Details

**INVCOV and CORR** For both INVCOV and CORR, we discretize the predicted continuous values to match the sparsity of the ground truth. The threshold is set to the $(1 - \frac{e}{n^2})$-th quantile of the predictions, where $e$ and $n$ represent the number of edges and nodes in the ground truth, respectively.

**FCI** We implement the Fast Causal Inference (FCI) algorithm using the *causal-learn* library[2]. FCI is a constraint-based causal discovery algorithm that identify causal relationships in the presence of latent confounders and selection bias. We use Fisher-Z test with $\alpha = 0.05$ significance Level during experiment.

**NOTEARS** We utilize the official implementation of the NOTEARS algorithm[3]. Following the default setting in the repository, we apply a threshold of $0.3$ to the estimated weight matrix to filter out weak edges before computing the Structural Hamming Distance (SHD).

**DiffAN** We implemented DiffAN by adopting the official hyperparameter configurations, which the original authors (Sanchez et al., 2023) noted are largely hard-coded and robust across diverse datasets. To strictly adhere to the non-approximated version of the algorithm, the `residue` parameter was set to True. For downstream evaluation requiring continuous scores such as AUC and mAP, we extracted the edge existence p-values and applied a $-\log_{10}$ transformation to derive the final confidence estimates. For metrics requiring a binary adjacency matrix such as SHD, we followed the hyperparameters $\alpha = 0.05$ as the threshold for edge pruning.

**SDCD** We utilized the official implementation of SDCD[4]. All hyperparameters followed the default settings provided in the official repository. We specified GPU as the computing device to ensure efficiency. To compute SHD, we apply a discretization threshold of 0.5.

**AVICI** Due to the high memory requirements encountered when attempting to train AVICI on our datasets (resulting in Out-of-Memory errors), we utilized the pre-trained checkpoints provided in the official repository[5]. For synthetic data, we employed the `scm-v0` model, which was pre-trained on diverse linear and non-linear datasets. For the SERGIO-GRN dataset, we utilized the `neurips-grn` checkpoint. All other settings remained consistent with our experimental setup, utilizing both interventional and observational data. We apply a discretization threshold of 0.5.

**SEA** For synthetic Stage 1 (10–100 nodes), we used SEA's publicly released checkpoint, as it was trained on data with the same distribution and generation code as ours. For the synthetic Stage 2 and SERGIO-GRN models, we train SEA with the same training data as ours. We use the GIES-based architecture. All training and testing configurations followed the default settings of SEA. We apply a discretization threshold of 0.5. SEA takes 48 GPU-hours to train in the synthetic Stage 2 and 72 GPU-hours in SERGIO-GRN.

---

[2]https://causal-learn.readthedocs.io
[3]https://github.com/xunzheng/notears
[4]https://github.com/azizilab/sdcd
[5]https://github.com/larslorch/avici

## D. `CauScale` Implementation Details

**Model Configuration.** The model consists of 10 layers with 128-dimensional embeddings and 16 attention heads. This configuration was determined through hyperparameter tuning across layers $\in \{8, 10\}$ and embedding dimensions $\in \{64, 128, 256\}$.

**Data Preprocessing.** Each set of data $\mathcal{D}$ is standardized variable-wise. For each variable $x_i$, we compute the normalized value via $\hat{x}_i = (x_i - \mu_i)/\sigma_i$, where $\mu_i$ represents the empirical mean and $\sigma_i$ is the standard deviation.

**Hardware Details** All training and inference tasks are conducted on NVIDIA H200 GPUs (141GB memory per GPU) and 164 CPU cores. Note that the baselines (e.g., AVICI) detailed in Section C are also evaluated in this environment.

**Training Strategy** We use the Adam optimizer with a learning rate of $1 \times 10^{-4}$. Training is performed on 8 GPUs using distributed data parallelism. For synthetic data, we adopt a two-stage training strategy. This design is motivated by two key factors: (1) It allows the model to capture fundamental causal relationships on simpler graphs (10–100 nodes) before generalizing to complex structures (up to 500 nodes). (2) Grouping graphs by size significantly reduces memory waste caused by excessive zero-padding when batching graphs of vastly different scales (e.g., mixing 10-node and 500-node graphs). Based on this, the training proceeds as follows:

- Stage 1 (10–100 nodes): Batch size of 8 for 37 hours.

- Stage 2 (150–500 nodes): Batch size of 1 for 2.75 hours.

For the SERGIO-GRN dataset, as the graph sizes vary within a narrower range (10–200 nodes), we train the model in a single stage with a batch size of 1 for 44 hours.

## E. Time and Memory Analysis

### E.1. Time Analysis

We provide a FLOP-level analysis of `CauScale`'s forward pass. Let $F$ denote the FFN inner dimension. LayerNorm, dropout, and softmax are omitted as their costs are negligible relative to linear projections and attention.

**Input embedding $C_{\text{embed}}$** Two parallel linear layers project the data stream and the graph prior:

$$C_{\text{embed}} = \underbrace{4mnd}_{\text{data: Linear}(2 \to d)} + \underbrace{2n^2 d}_{\text{prior: Linear}(1 \to d)}.$$

**Output head $C_{\text{head}}$** The prediction head applies two linear layers to $n(n-1)/2$ concatenated edge-pair embeddings of dimension $2d$:

$$C_{\text{head}} \approx 4n^2 d^2 + 6n^2 d.$$

**Axial-attention layer** An axial-attention layer on input $[m_b, n, d]$ performs column attention, row attention, and an FFN:

| Operation | Cost |
|---|---|
| Column attn: QKV + output projections | $8\,m_b n d^2$ |
| Column attn: scores + aggregation | $4\,m_b n^2 d$ |
| Row attn: QKV + output projections | $8\,m_b n d^2$ |
| Row attn: scores + aggregation | $4\,m_b^2 n d$ |
| FFN (width $F$) | $4\,m_b n F d$ |
| Total | $16\,m_b n d^2 + 4\,m_b n^2 d + 4\,m_b^2 n d + 4\,m_b n F d$ |

**Per-block cost** $C^{(b)}$   Each data-graph block at step $b$ contains three sub-layers:

*Data layer* — one axial-attention layer:

$$C^{(b)}_{\text{data}} = 16\,m_b n d^2 + 4\,m_b n^2 d + 4\,m_b^2 n d + 4\,m_b n F d.$$

*Data2graph layer* — axial-attention (FFN width $d$), two pooling heads, and one outer-product:

$$C^{(b)}_{\text{d2g}} = 20\,m_b n d^2 + 4\,m_b n^2 d + 4\,m_b^2 n d + 8\,n d^2 + 2\,n^2 d.$$

*Graph layer* — linear projection plus graph axial-attention (sequence length $n$):

$$C^{(b)}_{\text{graph}} \approx 18\,n^2 d^2 + 8\,n^3 d + 4\,n^2 F d.$$

Summing the three sub-layers:

$$C^{(b)} = 36\,m_b n d^2 + 8\,m_b n^2 d + 8\,m_b^2 n d + 4\,m_b n F d + 18\,n^2 d^2 + 8\,n^3 d + 2\,n^2 d + 4\,n^2 F d.$$

**Total cost** $C_{\textbf{total}}$   Let $S_1 = \sum_{b=0}^{B-1} m_b$ and $S_2 = \sum_{b=0}^{B-1} m_b^2$ denote the layer-wise sums. Then:

$$\begin{aligned}
C_{\text{total}} &= C_{\text{embed}} + C_{\text{head}} + \sum_{b=0}^{B-1} C^{(b)} \\
&= 36\,S_1 n d^2 + 8\,S_1 n^2 d + 8\,S_2 n d + 4\,S_1 n F d + (18B+4)\,n^2 d^2 + 8B\,n^3 d + 4B\,n^2 F d \\
&\quad + 2B\,n^2 d + 8Bn d^2 + 4mn d + 8n^2 d.
\end{aligned}$$

## E.2. Memory Analysis

**Proposition E.1** (Peak Activation Memory During Inference). *With the hierarchical reduction $m_b = m/r^{\lfloor b/k \rfloor}$, the peak GPU memory for activations during inference satisfies:*

$$M_{inference} = \Theta\big(mn(d+F) + Hm^2\big).$$

*Proof.* During inference, only the current block's live tensors occupy memory; earlier activations are freed. Since $m_b$ is non-increasing in $b$, the peak occurs at block $b = 0$ where $m_0 = m$. The dominant live tensors at block $b = 0$ are:

| Tensor | Lifetime | Shape | Elements |
|---|---|---|---|
| Sample tensor | persistent | $[m, n, d]$ | $mnd$ |
| Graph tensor | persistent | $[n, n, d]$ | $n^2 d$ |
| Sample FFN intermediate | transient | $[m, n, F]$ | $mnF$ |
| Graph FFN intermediate | transient | $[n, n, F]$ | $n^2 F$ |
| Sample col-attn weight | transient | $[H, n, n]$ | $Hn^2$ |
| Sample row-attn weight | transient | $[H, m, m]$ | $Hm^2$ |
| Graph row/col-attn weight | transient | $[H, n, n]$ | $Hn^2$ |

The persistent tensors are always live:

$$M_{\text{persistent}} = mnd + n^2 d = (m+n)nd.$$

The transient tensors are not simultaneously live; the peak transient cost is the maximum over the sub-operations within a block. Since $m > n$, the largest transient term is therefore:

$$M^{\text{max}}_{\text{transient}} = \max(mnF,\ Hm^2).$$

The total peak memory is:

$$M_{\text{inference}} = M_{\text{persistent}} + M_{\text{transient}}^{\text{max}} = (m+n)nd + \max(mnF, Hm^2).$$

Since $m > n$, we have $(m+n)nd = \Theta(mnd)$. Absorbing lower-order terms:

$$M_{\text{inference}} = \Theta\big(mn(d+F) + Hm^2\big). \qquad \square$$

## F. More results

### F.1. AID Metric Results

We report the performance under the AID metric family (Ancestor-AID, O-set-AID, Parent-AID) (Henckel et al., 2024) in Table 5 as a complement to Table 1. Lower AID values indicate better performance. AID metrics are formally defined only for DAGs and cannot be computed on cyclic graphs. To ensure a unified comparison, we apply the same cycle-breaking strategy as in Section 5.3, which iteratively detects cycles in the predicted graph and removes the edge with the lowest predicted score within each cycle, repeating until the graph is acyclic.

CauScale achieves the best AID in most of the settings, except for $n = 1000$ with sigmoid and polynomial, a challenging OOD setting in which both graph size and causal mechanism fall outside our training distribution (training graphs: $n \leq 500$, linear/NN mechanisms only). SDCD has the lowest score in these two cases. However, SDCD produces cyclic predictions for 100% of graphs at $n = 1000$ across all four mechanisms. At $n = 100$, SDCD also cycles 100% on 3/4 mechanisms (75% of graphs). Note that without our cycle-breaking procedure, SDCD's outputs at $n = 1000$ (100% cyclic) would not be valid inputs for AID computation at all.

### F.2. Error estimates

Below we provide full results of CauScale in Table 6 in addition to the main experiment in Table 1 with standard deviation as the error estimate (mean ± std). Each setting are conducted under 5 independent graphs.

### F.3. Effectiveness on observational datasets

We test the model for synthetic data in our main experiment (Section 5.3) on pure observational data; the result is shown below. This indicates that our model illustrates robustness without intervention data.

*Table 5.* AID metric results (lower is better). [†] indicates o.o.d. settings. Standard deviations are reported after $\pm$. For cyclic predictions, a confidence-guided greedy cycle-breaking procedure is applied before computing AID.

| | Synthetic (sample size = 1000) | | | | | |
|---|---|---|---|---|---|---|
| Model | Linear | | | NN non-add. | | |
| | Ancestor-AID | O-set-AID | Parent-AID | Ancestor-AID | O-set-AID | Parent-AID |
| *Setting:* $n = 100, \|E\| = 400$ | | | | | | |
| SEA-GIES | $0.174_{\pm 0.012}$ | $0.220_{\pm 0.013}$ | $0.464_{\pm 0.037}$ | $0.207_{\pm 0.018}$ | $0.243_{\pm 0.026}$ | $0.676_{\pm 0.040}$ |
| FCI | $0.232_{\pm 0.017}$ | $0.232_{\pm 0.016}$ | $0.737_{\pm 0.025}$ | $0.234_{\pm 0.019}$ | $0.234_{\pm 0.018}$ | $0.752_{\pm 0.044}$ |
| AVICI | $0.230_{\pm 0.012}$ | $0.231_{\pm 0.012}$ | $0.769_{\pm 0.034}$ | $0.227_{\pm 0.017}$ | $0.228_{\pm 0.016}$ | $0.713_{\pm 0.042}$ |
| SDCD | $0.228_{\pm 0.012}$ | $0.229_{\pm 0.011}$ | $0.749_{\pm 0.038}$ | $0.159_{\pm 0.031}$ | $0.223_{\pm 0.019}$ | $0.545_{\pm 0.044}$ |
| NOTEARS | $0.225_{\pm 0.013}$ | $0.230_{\pm 0.013}$ | $0.755_{\pm 0.025}$ | $0.227_{\pm 0.015}$ | $0.230_{\pm 0.015}$ | $0.743_{\pm 0.046}$ |
| CauScale (Ours) | $\mathbf{0.012}_{\pm 0.006}$ | $\mathbf{0.051}_{\pm 0.021}$ | $\mathbf{0.082}_{\pm 0.031}$ | $\mathbf{0.043}_{\pm 0.009}$ | $\mathbf{0.149}_{\pm 0.015}$ | $\mathbf{0.372}_{\pm 0.027}$ |
| *Setting:* $n = 1000, \|E\| = 2000$[†] | | | | | | |
| SEA-GIES | $0.050_{\pm 0.079}$ | $0.063_{\pm 0.099}$ | $0.121_{\pm 0.119}$ | $0.015_{\pm 0.007}$ | $0.020_{\pm 0.017}$ | $0.155_{\pm 0.024}$ |
| FCI | $0.010_{\pm 0.001}$ | $0.010_{\pm 0.001}$ | $0.083_{\pm 0.010}$ | $0.011_{\pm 0.001}$ | $0.011_{\pm 0.001}$ | $0.121_{\pm 0.013}$ |
| AVICI | $0.011_{\pm 0.001}$ | $0.011_{\pm 0.001}$ | $0.139_{\pm 0.015}$ | $0.012_{\pm 0.001}$ | $0.012_{\pm 0.001}$ | $0.144_{\pm 0.012}$ |
| SDCD | $0.010_{\pm 0.001}$ | $0.010_{\pm 0.001}$ | $0.065_{\pm 0.008}$ | $0.011_{\pm 0.001}$ | $0.012_{\pm 0.001}$ | $0.112_{\pm 0.004}$ |
| NOTEARS | $0.010_{\pm 0.001}$ | $0.010_{\pm 0.001}$ | $0.105_{\pm 0.014}$ | $0.011_{\pm 0.001}$ | $0.011_{\pm 0.001}$ | $0.127_{\pm 0.009}$ |
| CauScale (Ours) | $\mathbf{0.003}_{\pm 0.002}$ | $\mathbf{0.005}_{\pm 0.003}$ | $\mathbf{0.026}_{\pm 0.013}$ | $\mathbf{0.006}_{\pm 0.001}$ | $\mathbf{0.008}_{\pm 0.002}$ | $\mathbf{0.056}_{\pm 0.009}$ |

| | Synthetic (sample size = 1000, o.o.d. mechanisms) | | | | | |
|---|---|---|---|---|---|---|
| Model | Sigmoid[†] | | | Polynomial[†] | | |
| | Ancestor-AID | O-set-AID | Parent-AID | Ancestor-AID | O-set-AID | Parent-AID |
| *Setting:* $n = 100, \|E\| = 400$ | | | | | | |
| SEA-GIES | $0.217_{\pm 0.014}$ | $0.248_{\pm 0.018}$ | $0.656_{\pm 0.011}$ | $0.225_{\pm 0.012}$ | $0.228_{\pm 0.011}$ | $0.751_{\pm 0.038}$ |
| FCI | $0.240_{\pm 0.015}$ | $0.240_{\pm 0.015}$ | $0.754_{\pm 0.028}$ | $0.227_{\pm 0.012}$ | $0.227_{\pm 0.012}$ | $0.749_{\pm 0.040}$ |
| AVICI | $0.242_{\pm 0.015}$ | $0.242_{\pm 0.015}$ | $0.760_{\pm 0.027}$ | $0.228_{\pm 0.010}$ | $0.228_{\pm 0.010}$ | $0.800_{\pm 0.028}$ |
| SDCD | $0.227_{\pm 0.016}$ | $0.236_{\pm 0.015}$ | $0.703_{\pm 0.029}$ | $0.217_{\pm 0.012}$ | $0.219_{\pm 0.012}$ | $0.696_{\pm 0.042}$ |
| NOTEARS | $0.240_{\pm 0.016}$ | $0.241_{\pm 0.015}$ | $0.757_{\pm 0.033}$ | $0.226_{\pm 0.010}$ | $0.226_{\pm 0.010}$ | $0.748_{\pm 0.033}$ |
| CauScale (Ours) | $\mathbf{0.139}_{\pm 0.009}$ | $\mathbf{0.211}_{\pm 0.014}$ | $\mathbf{0.502}_{\pm 0.037}$ | $\mathbf{0.204}_{\pm 0.009}$ | $\mathbf{0.210}_{\pm 0.011}$ | $\mathbf{0.631}_{\pm 0.029}$ |
| *Setting:* $n = 1000, \|E\| = 2000$[†] | | | | | | |
| SEA-GIES | $0.011_{\pm 0.002}$ | $0.011_{\pm 0.002}$ | $0.136_{\pm 0.018}$ | $0.047_{\pm 0.072}$ | $0.062_{\pm 0.100}$ | $0.222_{\pm 0.131}$ |
| FCI | $0.011_{\pm 0.002}$ | $0.011_{\pm 0.002}$ | $0.133_{\pm 0.021}$ | $\mathbf{0.011}_{\pm 0.001}$ | $\mathbf{0.011}_{\pm 0.001}$ | $0.147_{\pm 0.008}$ |
| AVICI | $0.011_{\pm 0.002}$ | $0.011_{\pm 0.002}$ | $0.144_{\pm 0.022}$ | $0.012_{\pm 0.001}$ | $0.012_{\pm 0.001}$ | $0.148_{\pm 0.008}$ |
| SDCD | $\mathbf{0.010}_{\pm 0.002}$ | $\mathbf{0.010}_{\pm 0.002}$ | $\mathbf{0.120}_{\pm 0.020}$ | $\mathbf{0.011}_{\pm 0.001}$ | $\mathbf{0.011}_{\pm 0.001}$ | $\mathbf{0.143}_{\pm 0.006}$ |
| NOTEARS | $0.011_{\pm 0.002}$ | $0.011_{\pm 0.002}$ | $0.134_{\pm 0.020}$ | $\mathbf{0.011}_{\pm 0.001}$ | $\mathbf{0.011}_{\pm 0.001}$ | $0.145_{\pm 0.008}$ |
| CauScale (Ours) | $0.011_{\pm 0.002}$ | $0.013_{\pm 0.005}$ | $0.126_{\pm 0.023}$ | $0.030_{\pm 0.025}$ | $0.048_{\pm 0.047}$ | $0.208_{\pm 0.039}$ |

| | SERGIO-GRN (sample size = 20000) | | | | | |
|---|---|---|---|---|---|---|
| Model | $n = 100, \|E\| = 400$ | | | $n = 200, \|E\| = 400$ | | |
| | Ancestor-AID | O-set-AID | Parent-AID | Ancestor-AID | O-set-AID | Parent-AID |
| SEA-GIES | $0.233_{\pm 0.023}$ | $0.233_{\pm 0.023}$ | $0.792_{\pm 0.041}$ | $0.046_{\pm 0.010}$ | $0.046_{\pm 0.010}$ | $0.308_{\pm 0.056}$ |
| SDCD | $0.268_{\pm 0.025}$ | $0.271_{\pm 0.025}$ | $0.907_{\pm 0.045}$ | $0.091_{\pm 0.023}$ | $0.100_{\pm 0.022}$ | $0.602_{\pm 0.125}$ |
| NOTEARS | $0.236_{\pm 0.019}$ | $0.238_{\pm 0.019}$ | $0.802_{\pm 0.038}$ | $0.047_{\pm 0.009}$ | $0.048_{\pm 0.010}$ | $0.331_{\pm 0.061}$ |
| CauScale (Ours) | $\mathbf{0.188}_{\pm 0.033}$ | $\mathbf{0.220}_{\pm 0.025}$ | $\mathbf{0.643}_{\pm 0.073}$ | $\mathbf{0.042}_{\pm 0.008}$ | $\mathbf{0.042}_{\pm 0.008}$ | $\mathbf{0.257}_{\pm 0.037}$ |

*Table 6.* Full results of CauScale with standard deviation, in addition to Table 1. † indicates o.o.d. settings.

| **Synthetic** (sample size = 1000) | | | | |
|---|---|---|---|---|
| Mechanism | mAP (%) | SHD | AUC (%) | OA (%) |
| $n = 100, |E| = 400$ | | | | |
| Linear | $99.6 \pm 0.3$ | $15.2 \pm 7.0$ | $100.0 \pm 0.0$ | $99.9 \pm 0.1$ |
| NN | $89.0 \pm 0.5$ | $105.6 \pm 4.9$ | $98.5 \pm 0.1$ | $99.5 \pm 0.6$ |
| Sigmoid† | $84.4 \pm 2.1$ | $125.8 \pm 23.9$ | $94.9 \pm 0.9$ | $94.6 \pm 1.0$ |
| Polynomial† | $50.3 \pm 3.5$ | $252.2 \pm 16.8$ | $79.4 \pm 1.2$ | $81.7 \pm 2.8$ |
| $n = 1000, |E| = 2000$† | | | | |
| Linear | $96.6 \pm 4.1$ | $230.0 \pm 199.5$ | $99.9 \pm 0.1$ | $96.5 \pm 3.0$ |
| NN | $79.7 \pm 4.7$ | $835.0 \pm 381.3$ | $98.2 \pm 0.6$ | $96.6 \pm 0.5$ |
| Sigmoid† | $64.5 \pm 7.3$ | $1064.6 \pm 204.3$ | $95.3 \pm 0.9$ | $79.0 \pm 3.3$ |
| Polynomial† | $18.9 \pm 13.8$ | $3985.0 \pm 2975.4$ | $78.1 \pm 6.1$ | $59.7 \pm 8.6$ |

| **SERGIO-GRN** (sample size = 20000) | | | | |
|---|---|---|---|---|
| Graph Size | mAP (%) | SHD | AUC (%) | OA (%) |
| $n = 100, |E| = 400$ | $54.0 \pm 11.8$ | $290.8 \pm 60.3$ | $90.5 \pm 3.3$ | $94.2 \pm 1.9$ |
| $n = 200, |E| = 400$ | $39.2 \pm 11.9$ | $336.8 \pm 33.7$ | $90.8 \pm 2.7$ | $90.9 \pm 2.9$ |

*Table 7.* Model performance under pure observational data.

| Setting | Method | mAP | SHD | AUC | OA |
|---|---|---|---|---|---|
| $n = 100, |E| = 400$ | Linear | 98.3 | 39.8 | 99.9 | 99.0 |
| | NN non-add. | 86.0 | 133.6 | 98.1 | 98.8 |
| | Sigmoid | 76.4 | 155.8 | 90.9 | 88.9 |
| | Polynomial | 45.3 | 259.2 | 73.7 | 72.1 |
| $n = 1000, |E| = 2000$ | Linear | 97.7 | 168.0 | 100.0 | 96.9 |
| | NN non-add. | 71.8 | 1012.8 | 97.1 | 91.9 |
| | Sigmoid | 63.1 | 1033.6 | 95.2 | 77.6 |
| | Polynomial | 28.7 | 2074.2 | 82.0 | 63.7 |

