# OpenReview forum: "CauScale: Neural Causal Discovery at Scale"
_ICML.cc/2026/Conference — ICML 2026 regular_

### Official Review · Reviewer_grt9 · 2026-03-03

**Soundness:** 2
**Presentation:** 3
**Significance:** 3
**Originality:** 3
**Overall Recommendation:** 4
**Confidence:** 3

**Summary:**

The paper addresses the problem of scalable causal discovery under computational constraints. Specifically, they propos CauScale, a neural architecture for amortized causal discovery that scales to causal graphs with 1000 variables. The model first computes a graph prior which is the precision matrix of the input (which encodes conditional independence for Gaussian models). Both the features and the prior are encoded through pipelines of NNs called DataGraph blocks. The DataGraph combines the learnt embeddings from the features at the current layer with the embeddings of the graph at that layer and updates the graph embeddings accordingly. The DataGraph contains attention layers for three types of embeddings: the features, the graph, and the features that will update graph embeddings. These layers perform self-attention row-wise and column-wise on the current embeddings. The feature pipelines also goes through reduction layers to reduce the embedding size. This along with the tied attention formulation facilitates better memmory efficiency compared to competitive models. The experimental evaluation is conducted on both synthetic SCM-generated datasets and semi-synthetic single-cell gene regulatory network data. The synthetic benchmarks vary in graph size (up to 1000 nodes at test time), edge density, and functional mechanisms (linear, neural, sigmoid, polynomial), including out-of-distribution settings. On synthetic data with 100 nodes, CauScale achieves 99.6% mAP  on linear mechanisms and consistently outperforms baselines on nonlinear settings. When evaluated on 1000-node graphs the model maintains strong performance (e.g., 96.6% mAP on linear data) while reducing inference time compared to optimization-based methods. On SERGIO-based gene regulatory network data with large sample sizes, CauScale also achieves the best overall accuracy among the compared methods and remains memory-efficient in settings where some baselines run out of memory. Ablation studies show that the reduction unit and tied attention are crucial for scaling, and that removing the graph stream or the prior leads to noticeable performance degradation.

**Compliance With Llm Reviewing Policy:**

Affirmed.

**Final Justification:**

As evident by my answer to the authors, my main concern about cyclicity remains, and my other concerns are not addressed in the original body of the paper, so I propose this score.

**Key Questions For Authors:**

- How sensitive is the model to errors in the precision matrix prior, especially in small-sample or non-Gaussian settings?

- Does the model produce cyclic graphs in practice, and if so, how are such cases handled during evaluation?

- How does performance change when latent confounders are present, or when the data violates causal sufficiency assumptions?

**Limitations:**

Yes

**Strengths And Weaknesses:**

Strengths

- The paper directly targets scalability bottlenecks in amortized neural causal discovery with clear architectural interventions (reduction unit and tied attention).

- Strong empirical performance across graph sizes, including generalization to 1000-node graphs unseen during training.

- Comprehensive ablation studies that isolate the contribution of the reduction unit, graph stream, and attention design.

Weaknesses

- No explicit acyclicity enforcement at prediction time. How is the validity of the causal graphs ensured in this context?

- The model relies on the precision matrix as a structural prior.

- The evaluation relies on the SCM assumptions; robustness to confounding or model misspecification is unclear.

---

> ### Author Rebuttal · Authors · 2026-03-31
>
> We thank the reviewer for the thorough and constructive review. Below, we address each concern in detail.
> ## Weaknesses
> For W1, see response to Q2. For W2, see response to Q1. For W3, see response to Q3.
> ## Questions
> ### Q1: Sensitivity to prior
>
> The model is relatively robust to errors in the prior. The data stream independently processes raw observations without relying on the graph prior, and can compensate when the prior is noisy or misspecified.
>
> To directly test sensitivity in the small-sample regime, where the precision matrix is poorly estimated due to limited data, we conducted a controlled experiment varying the number of samples used to compute the prior (from 10 to 500), using the same model and settings as the main experiment, across graph sizes from n=10 to n=1000. Results are summarized in the table below:
>
> *mAP (%) across different sample sizes used to compute prior, averaged across 4 causal mechanisms:*
>
> |  | sample=10 | sample=100 | sample=200 | sample=500 |
> |---|---|---|---|---|
> | n=10, \|E\|=10 | 85.9 | 94.2 | 95.7 | 95.8 |
> | n=100, \|E\|=400 | 75.6 | 77.9 | 79.0 | 80.8 |
> | n=1000, \|E\|=2000 | 32.6 | 48.1 | 58.1 | 64.9 |
>
> Two patterns emerge: (1) the model remains robust to a noisy prior on most graphs, (2) the sensitivity to prior quality grows with graph size.
>
> As for non-Gaussian settings, unlike the small-sample setting — where we vary the number of samples used to compute the prior while keeping the data stream input fixed, yielding a clean ablation of prior quality alone — non-Gaussian sensitivity cannot be similarly isolated. Changing the noise distribution simultaneously affects both the precision matrix and the raw observations fed into the data stream, making it impossible to attribute performance changes specifically to prior inaccuracies. We therefore acknowledge this as an open question.
>
> ### Q2: Cyclicity
>
> We report the cyclicity in the main experiment and additionally test on n=500, the largest graph size seen during training:
>
> *Synthetic*:
>
> | Setting | Linear | NN | Sigmoid (ood) | Polynomial (ood) |
> |---------|--------|----|---------------|------------------|
> | n=100, \|E\|=400 | 0 | 0 | 0 | 0 |
> | n=500, \|E\|=1000 | 0 | 0 | 0 | 0 |
> | n=1000, \|E\|=2000 (out-of-distribution) | 0.2 | 0.4 | 0.6 | 1 |
>
> *SERGIO-GRN*:
>
> | Setting | cyclicity |
> |---------|-------------|
> | n=100, \|E\|=400 | 0 |
> | n=200, \|E\|=400 | 0 |
>
> The model produces zero cycles across all in-distribution graph size settings. In the out-of-distribution graph size setting (synthetic n=1000, 2000 edges), cycles do occur, though expected given the graph is 2 times larger than seen during training.
>
> As for how cyclic cases are handled during evaluation, all metrics we report (SHD, mAP, AUC, and OA) are defined directly on the predicted edge set and do not require the graph to be acyclic, so no special handling is needed.
> Nevertheless, to further address potential concerns about acyclicity guarantees, we implement a cycle-breaking strategy that iteratively removes the edge with the lowest model confidence score in each detected cycle. We report SHD and the average number of edges removed per graph before and after applying the procedure:
>
> *Synthetic (n=1000):*
>
> | Mechanism | SHD (original) | SHD (after cycle-breaking) | Edges removed (avg. per graph) |
> |-----------|---------------|---------------------------|-------------------------------|
> | Linear | 230 | 230 | 0.4 |
> | NN | 835 | 834.6 | 0.6 |
> | Sigmoid (ood) | 1064.6 | 1064.4 | 0.6 |
> | Polynomial (ood) | 3985 | 3891.8 | 103 |
>
> Applying this cycle-breaking procedure consistently improves or preserves SHD.
>
> ### Q3: Robustness to latent confounders
>
> We simulate latent confounding via selective node removal. Specifically, we hide the top-$k$ nodes ranked by out-degree, as high out-degree nodes act as strong common causes. After removal, the observable sub-graph contains $n_{\text{obs}} = 100 - k$ nodes. We vary
> $k$ as  1, 2, 4, 8, 16, 32, 64  to cover a wide range of confounding severity.
> The table below reports mAP (%) on the observable sub-DAG for $n=100$ node graphs:
>
> | $k$ (hidden) | $n_{\text{obs}}$ | Linear | NN | Sigmoid | Polynomial |
> |---|---|---|---|---|---|
> | 0 | 100 | 99.6 | 89.0 | 84.4 | 50.3 |
> | 1 | 99 | 99.1 | 88.1 | 82.9 | 49.7 |
> | 2 | 98 | 98.8 | 87.1 | 81.6 | 48.5 |
> | 4 | 96 | 97.9 | 86.6 | 79.5 | 45.6 |
> | 8 | 92 | 97.1 | 84.0 | 76.1 | 39.8 |
> | 16 | 84 | 94.5 | 80.4 | 71.3 | 33.9 |
> | 32 | 68 | 88.4 | 71.9 | 60.7 | 24.5 |
> | 64 | 36 | 68.3 | 46.7 | 43.4 | 7.9 |
>
> Under mild confounding ($k \leq 8$), the model remains robust across all mechanisms, with linear mAP staying above 97%. Under severe confounding ($k = 64$, where 64% of nodes are hidden), performance drops substantially, which is expected given that the observable sub-graph loses much of its causal structure. Overall, the model remains reliable under moderate confounding.

---

> > ### Author Rebuttal · Reviewer_grt9 · 2026-04-02
> >
> > I thank the reviewers for their answers, however my main concerns persist. The sensitivity analysis to the precision matrix prior is helpful, but it does not fully isolate the effect of prior quality, as changes in graph size simultaneously affect multiple factors. A more controlled analysis, e.g., directly injecting noise into the precision matrix, would provide a clearer understanding of the model’s robustness to prior misspecification. The discussion on cyclicity clarifies that cycles are rare in-distribution, but the lack of an explicit acyclicity constraint leads to cycles in larger, out-of-distribution graphs, which are precisely the challenging settings of interest. The additional experiments on latent confounding are informative but still highlight a degradation in performance under stronger violations of causal assumptions. Overall, while the rebuttal strengthens the analysis, it does not fully address my concerns and I therefore maintain my original evaluation.

---

> > > ### Author Response · Authors · 2026-04-03
> > >
> > > We thank the reviewer for the follow-up comments and address each point below.
> > >
> > > **(1) Sensitivity analysis to precision matrix**: We respectfully note a potential misreading of our experimental design. In the sensitivity analysis, graph size is held fixed within each row — the manipulation variable is solely the number of samples used to compute the precision matrix prior.
> > > Within any fixed graph size, varying the number of samples has a direct effect only on the precision matrix prior — all other factors are affected only through the matrix prior's influence. Any change in model output can therefore be attributed to prior quality only. Hence, this design has reliably justified the robustness of CauScale to prior misspecification.
> > >
> > > Following the reviewer's suggestion, we further conduct an experiment directly injecting additive symmetric Gaussian noise into the precision matrix. Specifically, we generate a random matrix $Z \in \mathbb{R}^{n \times n}$ with $Z_{ij} \overset{\text{i.i.d.}}{\sim} \mathcal{N}(0, 1)$, symmetrize it as $\tilde{Z} = (Z + Z^\top) / 2$, and inject $\alpha \cdot \text{std}(\rho) \cdot \tilde{Z}$ into $\rho$, where $\rho$ is the clean estimated precision matrix used in the main experiment.
> > > $\alpha$ measures noise level relative to the precision matrix's own scale. Performance of CauScale across different noise levels is reported below (mAP %, averaged across 4 causal mechanisms):
> > >
> > > | Graph size | $\alpha=0.0$ | $\alpha=0.2$ | $\alpha=0.4$ | $\alpha=0.6$ | $\alpha=0.8$ |
> > > |:---|---:|---:|---:|---:|---:|
> > > | n=10, \|E\|=10 | 95.8 | 94.5 | 91.9 | 89.7 | 88.1 |
> > > | n=100, \|E\|=400 | 80.8 | 80.0 | 78.9 | 77.9 | 76.9 |
> > > | n=1000, \|E\|=2000 | 64.9 | 63.3 | 62.3 | 61.7 | 59.3 |
> > >
> > > The results show that even at $\alpha=0.8$ (where the noise standard deviation is 80% of the precision matrix's own standard deviation, already a severe misspecification), mAP only drops by at most 7.7%. These results confirm that CauScale is robust to prior misspecification in the precision matrix.
> > >
> > > **(2) Cyclicity Challenge:**
> > > We agree that acyclicity is an important consideration. We believe the post-processing step we introduced already addresses this concern: it iteratively removes the edge with the lowest confidence score in each detected cycle, and the results show that cycles produced (if any) are sparse and can be corrected easily without performance degradation. Furthermore, it is worth pointing out that cycles are not inherently pathological — real-world causal systems may exhibit feedback loops, and assuming strict acyclicity can be an error in many practical settings.
> > >
> > > **(3) Degradation under stronger confounding severity:**
> > > We would like to highlight that the strong confounding setting (stronger violations of causal assumptions) evaluated removes 64% of nodes, which represents an extremely difficult scenario that would severely challenge any method. Performance degradation under such extreme violation of causal assumptions is expected. To support this claim, we benchmark CauScale against the baselines in our experiment under the same $k$=64 confounding condition. Results are reported as mAP (%), where ood indicates out-of-distribution mechanisms:
> > >
> > > | Method | Linear | NN | Sigmoid (ood) | Polynomial (ood) | Avg |
> > > |:---|---:|---:|---:|---:|---:|
> > > | FCI | 4.0 | 3.3 | 2.4 | 2.2 | 3.0 |
> > > | NOTEARS | 20.4 | 7.3 | 5.4 | 2.2 | 8.8 |
> > > | SDCD | 31.5 | 24.1 | 18.4 | **12.0** | 21.5 |
> > > | AVICI | 22.2 | 15.5 | 20.1 | 2.3 | 15.0 |
> > > | SEA | 47.3 | 18.9 | 30.4 | 5.5 | 25.5 |
> > > | **CauScale (Ours)** | **68.3** | **46.7** | **43.4** | 7.9 | **41.6** |
> > >
> > > CauScale achieves the highest average mAP (41.6%) under extreme confounding, leading across most causal mechanisms. This confirms its robustness and advantage even under strong violations of causal assumptions.

---

### Official Review · Reviewer_nrid · 2026-03-11

**Soundness:** 3
**Presentation:** 2
**Significance:** 3
**Originality:** 3
**Overall Recommendation:** 4
**Confidence:** 3

**Summary:**

This paper proposes CauScale, an efficient neural architecture for causal discovery. The model employs a two-stream design, including a data stream and a graph stream, and improves time and space efficiency through reduction units and tied attention weights. Experiments on synthetic and real datasets demonstrate the effectiveness and efficiency of this approach.

**Compliance With Llm Reviewing Policy:**

Affirmed.

**Key Questions For Authors:**

1. Referring to Figure 3, why is the mAP higher when using pooling with 100 points than when not using pooling?
2. You mentioned "We retrain the two ablation versions on our synthetic train set," but which one specifically? The data reported in the table does not match the bar chart (100, 400).

**Limitations:**

yes

**Strengths And Weaknesses:**

Strong Points
1. The experimental results are excellent, showing significant improvements over the baseline across multiple accuracy metrics, while also demonstrating a clear advantage in efficiency.
2. The experiments are comprehensive, including multiple ablation experiments, hyperparameter analysis, and comparative experiments on pooling methods.
3. The paper is well-written, with accurate and clear framework diagrams, and the methods section uses abundant and concise formulas.

Weak Points
1. When comparing spatiotemporal efficiency, it is recommended to select the optimal baseline and explain its spatiotemporal efficiency using formulas to provide a more intuitive comparison.
2. The OA values in Table 1 SDCD should be bolded; and the format should be standardized, as some numbers are 100 and others are 100.0, ensuring consistency in significant figures or decimal places. Also, some figures that only report percentages should be explained, such as mAP; and the meaning of the '-' symbols in the table needs to be explained.

In summary, although this paper has some issues with the details of the experimental results reporting, it is technically rigorous, explores important issues, and designs a novel architecture to achieve efficient and accurate causal discovery. It is a good paper.

---

> ### Author Rebuttal · Authors · 2026-03-31
>
> We thank the reviewer for the careful reading and the constructive suggestions. Below we respond to each point in detail.
> ## Weaknesses
> ### W1: Comparing spatiotemporal efficiency with optimal baseline
>
> We select AVICI as the primary baseline for a formula-based spatiotemporal efficiency comparison, as it is the closest baseline to CauScale in both paradigm (amortized, pre-training-based) and architectural design. Throughout, $m$ denotes the
> number of observational samples, $n$ the number of nodes, and $H$ the number of
> attention heads.
>
> **Time Efficiency**
>
> AVICI applies no compression along the sample dimension, so its per-block computational costs
> for sample-axis and node-axis attention are:
>
> $$C_{\text{sample}}^{\text{AVICI}} \propto nm^2, \quad C_{\text{node}}^{\text{AVICI}} \propto mn^2$$
>
> CauScale introduces the reduction unit, which reduces the sample dimension by a factor of
> $r$ every $k$ blocks. The average per-block cost becomes:
>
> $$C_{\text{sample}}^{\text{Ours}} \propto \frac{nm^2}{B} \sum_{b=0}^{B-1} r^{-2\lfloor b/k \rfloor}, \quad C_{\text{node}}^{\text{Ours}} \propto \frac{n^2m}{B} \sum_{b=0}^{B-1} r^{-\lfloor b/k \rfloor}$$
>
> **Space Efficiency**
>
> AVICI uses standard self-attention and stores axis-specific attention maps. For sample-axis and node-axis attention, the memory costs are:
>
> $$M_{\text{sample}}^{\text{AVICI}} = O(Hnm^2), \quad  M_{\text{node}}^{\text{AVICI}} = O(Hmn^2)$$
>
> CauScale adopts tied attention weights, sharing attention weights across
> the target axis and storing only:
>
> $$M_{\text{sample}}^{\text{Ours}} = O(Hm^2), \quad  M_{\text{node}}^{\text{Ours}} = O(Hn^2)$$
>
> ### W2: Format standardization and meaning of '-' symbols
>
> We will standardize the formatting in the revision accordingly.
> Regarding the '−' symbol: it indicates that OA is not applicable, as CORR and INVCOV
> output symmetric matrices and thus provide no edge orientation. We will add a footnote to clarify this.
>
> ## Questions
> ### Q1: Higher mAP with pooling
>
> We attribute this to the training efficiency brought by the reduction unit. Without it, the network must process all $m$ samples throughout every layer, leading to higher training complexity and potentially degraded optimization. The reduction unit compresses redundant or noisy signals into more compact and informative representations, progressively reducing the computational burden and allowing the network to learn more effective representations. This is consistent with Figure 7, where mAP peaks at a moderate sample size and degrades beyond it, suggesting that an excessive number of samples can overwhelm the network's representational capacity and hurt performance.
>
> ### Q2: Clarification on ablation training set and data discrepancy
>
> The ablation models are trained on the same synthetic training set as the main
> experiments, with results averaged across all four causal mechanisms (linear, NN,
> sigmoid, and polynomial).
>
> Regarding the discrepancy: given the consistently small variance observed across the main experiments, the bar chart in Figure 4 was generated using a single graph instance per configuration as a lightweight check. We acknowledge this may cause minor numerical differences. To address this, we re-ran the evaluation over 5 independent graph instances and report the averaged results with standard deviations below. The updated figure, regenerated using the same plotting code, shows negligible visual differences from the original, and the conclusions remain fully consistent.
>
> *Advantage of data-graph block over the block containing the data layer only (mAP %):*
>
> | Graph Size | CauScale (Ours) | w/o Graph Prior | w/o Graph Stream |
> |:-----------|:----------------|:----------------|:-----------------|
> | n=10, \|E\|=10 | 95.9 ± 2.1 | 84.5 ± 5.3 | 54.2 ± 5.7 |
> | n=20, \|E\|=80 | 88.4 ± 1.0 | 76.7 ± 1.3 | 51.1 ± 2.4 |
> | n=50, \|E\|=200 | 83.4 ± 0.9 | 77.2 ± 1.1 | 50.9 ± 2.1 |
> | n=100, \|E\|=400 | 80.8 ± 1.0 | 76.5 ± 1.4 | 60.7 ± 1.2 |
> | n=1000, \|E\|=2000 | 64.9 ± 4.2 | 49.8 ± 7.7 | 3.6 ± 0.7 |

---

> > ### Author Rebuttal · Reviewer_nrid · 2026-04-07
> >
> > I thank the authors for their detailed response to the concerns I raised, but I will maintain my original score.

---

### Official Review · Reviewer_u8Ys · 2026-03-12

**Soundness:** 2
**Presentation:** 2
**Significance:** 2
**Originality:** 3
**Overall Recommendation:** 5
**Confidence:** 3

**Summary:**

The authors aim to address the widely recognized scaling issues in causal discovery, particularly around the number of system variables and requisite temporal history. They propose a neural architecture - dubbed CauScale - to address these issues. While not explicitly stated in their manuscript, CauScale appears to be an amortized causal graph discovery technique which the authors compare against several non-amortized and amortized baselines.

**Compliance With Llm Reviewing Policy:**

Affirmed.

**Final Justification:**

The authors have provided extensive quantitative evidence and agreed to ammend gaps in the description of their method and related methods during the rebuttal period. Provided they make the changes discussed during the rebuttal phase and include results shared with the reviewers, I see no strong reasons why this paper should not be accepted into the conference.

**Key Questions For Authors:**

Q1: Can/Will you update your reported results with a description of how many unique graphs/systems were involved in your experiments and the associated error estimates on performance?

      Explanation of how possible responses would change my evaluation of the paper: implementing these two updates could improve my assessment of soundness and significance.

Q2: Can/Will you update your discussion around how your proposed method relates to existing literature (esp. with regards to the Granger causal methods mentioned in “Strengths and Weaknesses” and an explicit declaration of whether your model is amortized or not)?

      Explanation of how possible responses would change my evaluation of the paper: improved discussion around how this method relates to existing literature could improve my assessment of presentation.

Q3: Can/Will you include evaluations using the AID family of metrics (discussed in “Strengths and Weaknesses”) to assess how the CauScale technique captures the presence of causal effects (not just direct connections)?

      Explanation of how possible responses would change my evaluation of the paper: inclusion of these results could improve my assessment of soundness, presentation, and significance.

**Limitations:**

Yes

**Strengths And Weaknesses:**

Soundness:

 i) Is the submission technically sound?

      Response: Yes, for the most part; the authors take care to avoid compressing systems along the variable/node axes (which would have hurt interpretability), and do not assume acyclicity, which their compressions along the temporal axes would pre-empt (though this aspect of their method is not discussed, and the authors give different reasoning for rejecting the acyclicity assumption). There is precedent in published literature for relaxing acyclicity constraints, particularly in neurological research (one of the applications explored in the paper).

 ii) Are claims well supported (e.g., by theoretical analysis or experimental results)?

      Response: The experimental results presented are not detailed enough to draw a conclusion regarding the claims being made. Specifically, it is unclear how many graphs are actually used in experiments, and there are no error estimates reported across experiments. Together, these gaps raise the concern that the authors are trying to discover one graph per hyperparameter setting, which would be insufficient empirical evidence to properly evaluate even non-amortized algorithms. There are no significant theoretical claims made, and as mentioned above, the relevant theory for their method does appear to be properly applied.

 iii) Are the methods used appropriate?

      Response: As alluded to, the experimental methods used are insufficient. In addition to the need for error estimates around model performance, the metrics explored by the authors - “Structural Hamming Distance (SHD), Mean Average Precision (mAP), Area Under the ROC Curve (AUC), and Orientation Accuracy (OA)” - focus entirely on ‘one-shot’ edge prediction, and do not give a sense of ‘functional connectivity’ in the predicted graphs. Metrics that account for causal chaining (i.e. A causes B causes C implies that A causes C via B) should be included for a more comprehensive analysis. I recommend the authors look into/use the metrics used by (Henckel et al 2024) in which they discuss the differences between estimating ‘direct causal relationships’ (measured by SHD) and estimating the presence of ‘causal effects’ (measured by the AID family of metrics).

      Ref: Henckel, Leonard, Theo Würtzen, and Sebastian Weichwald. "Adjustment identification distance: A gadjid for causal structure learning." arXiv preprint arXiv:2402.08616 (2024).

 iv) If the paper includes theoretical results, are the proofs correct and based on reasonable assumptions?

      Response: N/A

 v) If the paper includes empirical results, are the experiments well-designed?

      Response: Again, this is unclear. The paper does not detail how many distinct causal graphs are used in their evaluations.

 vi) Are the authors careful and honest about evaluating both the strengths and weaknesses of their work?

      Response: The authors appear to present an honest evaluation of the method, as evidenced by the less-than-perfect (albeit still promising) results in Table 1.

Presentation (Note that a superbly written paper provides enough information for an expert reader to reproduce its results.):

 i) Is the submission clearly written and well structured? (If not, please make constructive suggestions for improving its clarity.)

      Response: The submission requires a few significant improvements regarding clarity. To improve clarity, the authors need to explicitly state whether their algorithm is amortized or not, how many distinct causal graphs were used in their evaluations, and their stance on the acyclical bias in their results. On this last point regarding acyclicity: the authors acknowledge their method (like prior methods in literature) cannot assume acyclicity in causal graphs, yet they seem to go to great lengths to demonstrate that their method produces acyclical predictions at a higher rate than baselines (e.g. in Figure 5 and the final sentences of Section 4.5 and the “Attention Shape and Output Head” paragraph) and they do not discuss this apparent gap between what they do assume (cycles are possible) and what they want to demonstrate (acyclicity).

 ii) Is the overall narrative easy to follow?

      Response: Yes, the paper is well organized in this way.

 iii) Does the work properly position itself in the context of prior/concurrent literature and clearly discuss how it differs?

      Response: No. In addition to not explicitly stating where their proposed method falls in the broader literature (they need to state whether it is amortized or not), the authors do not discuss Granger causality (which directly relates to their ‘observational setting’), and thus neglect to cite/compare against relevant work done in leveraging Granger causal-concepts for causal discovery and hypotheses generation (Gallagher et al 2017, Tank et al 2021, and Brown et al 2025).

      Refs:

            Gallagher, Neil, et al. "Cross-spectral factor analysis." Advances in neural information processing systems 30 (2017).

            Tank, Alex, et al. "Neural granger causality." IEEE Transactions on Pattern Analysis and Machine Intelligence 44.8 (2021): 4267-4279.

            Brown, Zachary C., and David Carlson. "Generating Hypotheses of Dynamic Causal Graphs in Neuroscience: Leveraging Generative Factor Models of Observed Time Series." International Conference on Machine Learning. PMLR, 2025.

Significance:

 i) Does the paper address an important or relevant problem?

      Response: Yes (scaling in causal discovery).

 ii) Does it advance understanding, capabilities, or practice in machine learning?

      Response: Yes, if the empirical findings are statistically significant (which it is not yet clear that they are).

 iii) Could it influence future research or applications (e.g., other researchers or practitioners are likely to use the ideas or build on them)?

      Response: Yes, particularly as an analysis tool in the broader scientific community.

 iv) Is the scope of impact broad or specialized, and is that appropriate for the contribution?

      Response: There is the potential for broad impact, which would be appropriate.

 v) Even if the improvements are modest or domain-specific, could they unlock new directions or provide practical utility?

      Response: Yes.

Originality:

 i) Does the work provide new insights, deepen understanding, or highlight important properties of existing methods?

      Response: Marginally.

 ii) Does the work introduce new tasks, methods, theory, data, or perspectives that advance the field in some dimensions?

      Response: Yes (methods).

 iii) Does this work offer a novel combination of existing techniques, and is the reasoning behind this combination well-articulated?

      Response: Yes it offers a novel combination of existing techniques (it introduces a variant of the popular neural network paradigm), and the reasoning for certain choices is well articulated.

 iv) Are the contributions clearly distinguished from closely related literature, and is the novelty well justified?

      Response: Yes, I would say so.

---

> ### Author Rebuttal · Authors · 2026-03-31
>
> We thank the reviewer for the thorough and constructive feedback.
>
> ## Weaknesses
>
> Regarding **Presentation i)**,
> we clarify our treatment of acyclicity as follows. We deliberately do not enforce acyclicity as a hard constraint, for two reasons: (1) real-world data may contain cycles, and (2) explicit DAG enforcement incurs computational overhead. However, we do report low cyclicity as an empirical observation because, in the predominant benchmark setting where ground-truth graphs are DAGs, producing fewer cycles is a practically desirable property that reflects better edge orientation.
>
> Other points raised under Weaknesses are addressed in the responses to Q1–Q3 below.
>
> ## Questions
> ### Q1: Graphs per settings and error estimates
> In Section 5.2 (line 267–269), we state that 5 independent graphs are used per setting, consistent with prior work (e.g., SEA, SDCD). Below we provide full results of CauScale with standard deviation as the error estimate (mean ± std).
>
> *Synthetic, n=100, |E|=400:*
>
> | Mechanism | mAP (%) | SHD | AUC (%) | OA (%) |
> |---|---|---|---|---|
> | Linear | 99.6±0.3 | 15.2±7.0 | 100.0±0.0 | 99.9±0.1 |
> | NN | 89.0±0.5 | 105.6±4.9 | 98.5±0.1 | 99.5±0.6 |
> | Sigmoid (ood) | 84.4±2.1 | 125.8±23.9 | 94.9±0.9 | 94.6±1.0 |
> | Polynomial (ood) | 50.3±3.5 | 252.2±16.8 | 79.4±1.2 | 81.7±2.8 |
>
> *Synthetic, n=1000, |E|=2000 (ood setting):*
>
> | Mechanism | mAP (%) | SHD | AUC (%) | OA (%) |
> |---|---|---|---|---|
> | Linear | 96.6±4.1 | 230.0±199.5 | 99.9±0.1 | 96.5±3.0 |
> | NN | 79.7±4.7 | 835.0±381.3 | 98.2±0.6 | 96.6±0.5 |
> | Sigmoid (ood) | 64.5±7.3 | 1064.6±204.3 | 95.3±0.9 | 79.0±3.3 |
> | Polynomial (ood) | 18.9±13.8 | 3985.0±2975.4 | 78.1±6.1 | 59.7±8.6 |
>
> *SERGIO-GRN:*
>
> | Graph Size | mAP (%) | SHD | AUC (%) | OA (%) |
> |---|---|---|---|---|
> | n=100, \|E\|=400 | 71.4±11.8 | 290.8±60.3 | 95.1±3.3 | 94.2±1.9 |
> | n=200, \|E\|=400 | 34.5±11.9 | 336.8±33.7 | 90.3±2.7 | 90.9±2.9 |
>
> ### Q2: Clarification on amortized inference and relation to Granger causal methods
> (1) CauScale is an amortized method: it is pre-trained once and performs zero-shot inference at test time. (2)The Granger-causal methods are designed for time-series data, where causality is inferred from temporal precedence. CauScale, by contrast, targets the i.i.d. observational-plus-interventional setting, where samples are exchangeable and there is no temporal ordering to exploit. Consequently, Granger methods cannot be directly applied to our benchmarks. We will add a paragraph in the Related Work section citing the three references above and clarifying this distinction.
>
> ### Q3: AID metrics
>
> We evaluate the AID metric family (Ancestor-AID, O-set-AID, Parent-AID) across all baselines in main experiment (Table 1).
> As AID metrics are formally defined only for DAGs, for any cyclic prediction, we apply a greedy cycle-breaking procedure which iteratively detects cycles via DFS and removes the lowest-confidence edge, until the graph is acyclic. Results on n=100 are shown below. CauScale achieves the best AID across all mechanisms.
>
> *Synthetic, n=100, |E|=400:*
> | Method | Linear Anc-AID | Linear Oset-AID | Linear Par-AID | NN Anc-AID | NN Oset-AID | NN Par-AID | Sigmoid Anc-AID | Sigmoid Oset-AID | Sigmoid Par-AID | Poly. Anc-AID | Poly. Oset-AID | Poly. Par-AID |
> | :--- | :---: | :---: | :---: | :---: | :---: | :---: | :---: | :---: | :---: | :---: | :---: | :---: |
> | SEA-GIES | 0.174 | 0.220 | 0.464 | 0.207 | 0.243 | 0.676 | 0.217 | 0.248 | 0.656 | 0.225 | 0.228 | 0.751 |
> | FCI | 0.232 | 0.232 | 0.737 | 0.234 | 0.234 | 0.752 | 0.240 | 0.240 | 0.754 | 0.227 | 0.227 | 0.749 |
> | AVICI | 0.230 | 0.231 | 0.769 | 0.227 | 0.228 | 0.713 | 0.242 | 0.242 | 0.760 | 0.228 | 0.228 | 0.800 |
> | SDCD | 0.228 | 0.229 | 0.749 | 0.159 | 0.223 | 0.545 | 0.227 | 0.236 | 0.703 | 0.217 | 0.219 | 0.696 |
> | NOTEARS | 0.225 | 0.230 | 0.755 | 0.227 | 0.230 | 0.743 | 0.240 | 0.241 | 0.757 | 0.226 | 0.226 | 0.748 |
> | CauScale (Ours) | **0.012** | **0.051** | **0.082** | **0.043** | **0.149** | **0.372** | **0.139** | **0.211** | **0.502** | **0.204** | **0.210** | **0.631** |
>
> We also compute AID in the main experiment on synthetic data with n=1000 and SERGIO-GRN. On synthetic n=1000 (ood graph size), CauScale achieves the best AID on linear and NN mechanisms. On sigmoid/polynomial (ood mechanism), SDCD obtains better AID. However, SDCD produces cyclic outputs for 100% of graphs at n=1000 across all four mechanisms, meaning it fails to produce even one valid DAG without our cycle-breaking post-processing.
> On SERGIO-GRN, CauScale achieves the best AID at both n=100 and n=200. *Standard deviations and full AID tables for synthetic n=1000 and SERGIO-GRN are omitted due to space constraints and available upon request.*

---

> > ### Author Rebuttal · Reviewer_u8Ys · 2026-04-01
> >
> > I thank the authors for their detailed response to the concerns I raised. In addition to the changes mentioned in the Q2 response, will the authors agree to include the tables/results included in their responses to Q1 and Q3 in the appendices of the final paper (they were not explicit on this point in their response)?

---

> > > ### Author Response · Authors · 2026-04-01
> > >
> > > Yes, we will include all tables and results from our responses to Q1 and Q3 in the appendices of the final paper. Thank you for raising this point.

---

### Official Review · Reviewer_1vci · 2026-03-12

**Soundness:** 3
**Presentation:** 4
**Significance:** 3
**Originality:** 3
**Overall Recommendation:** 5
**Confidence:** 4

**Summary:**

This paper proposes CauScale, an amortized neural architecture for scalable causal discovery on large graphs. The main goal is to improve both time efficiency and memory efficiency relative to prior neural and optimization-based methods, especially in regimes with hundreds to thousands of variables. The model uses a two-stream design: a data stream processes observational/interventional data, while a graph stream incorporates a statistical graph prior and structural messages distilled from the data stream. To improve scalability, the method introduces a reduction unit that compresses the sample dimension during processing and uses tied attention weights to reduce attention memory. The paper evaluates CauScale against classical and recent baselines including FCI, NOTEARS, SDCD, DiffAN, AVICI, and SEA on synthetic SCM data and SERGIO-based semi-synthetic gene regulatory network data. Across the reported settings, CauScale achieves the best or near-best discovery metrics while also substantially improving inference speed, including reported speedups of 4× to 13,000× over prior methods and successful training up to 500 nodes with inference up to 1000 nodes. Overall, the paper’s contribution is an engineering and architectural advance for amortized large-scale causal discovery, with a clear focus on practical scalability rather than new identifiability theory.

**Compliance With Llm Reviewing Policy:**

Affirmed.

**Final Justification:**

This rebuttal has addressed my main concern

**Key Questions For Authors:**

1. Does the reported inference time include graph prior computation or only neural inference?
2. How sensitive is the model to inaccuracies in the inverse covariance prior?
3. What is the total pretraining cost of CauScale compared to AVICI or SEA?
4. How frequently does the model output cyclic graphs in the main experiments?
5. How does performance change in purely observational settings without intervention indicators?
6. How does the model behave in the presence of latent confounders?

**Limitations:**

1. The method relies heavily on simulator-generated training data and may inherit simulator bias.
2. Most evidence comes from synthetic or semi-synthetic experiments.
3. The framework assumes causal sufficiency and structured interventions.
4. Acyclicity is not strictly enforced during prediction.
5. Robustness to mechanism and noise distribution shifts remains limited.

**Strengths And Weaknesses:**

Strengths

1. The paper addresses a real bottleneck in causal discovery: many existing methods become impractical on large graphs due to expensive optimization, conditional independence testing, or large attention tensors. The motivation for a scalable amortized architecture is well justified.

2. The architectural ideas are coherent and reasonably well aligned with the bottlenecks.
The three main design choices are i) a reduction unit to shrink the observation dimension over depth, ii) tied attention weights to reduce memory, and iii) a data-graph block to pass distilled relational evidence from the data stream into the graph stream.
These design choices are internally consistent with the stated objective of reducing time and space cost while preserving structural signal.

3.  The paper includes ablations for: i) with/without reduction unit, ii) with/without graph prior, iii) with/without graph stream,
iv) tied attention vs vanilla attention, v) prediction head variants, vi) pooling strategy in the reduction unit.
These help support the claim that the proposed system is not just a bundle of arbitrary design choices.

4. The paper reports both inference time and peak GPU memory, and shows clear practical gains over prior methods. This is especially important because scalability is the central contribution.



Weaknesses

1.  The main novelty is architectural efficiency rather than a new causal principle, identifiability argument, or theoretical guarantee. The paper improves amortized discovery infrastructure, but it does not substantially deepen understanding of when the learned model succeeds or fails from a causal/statistical perspective. This is acceptable, but it limits the conceptual novelty.

2. The experiments use SCM-generated synthetic data and SERGIO-based GRN data. While this is standard in the literature, it means the paper does not demonstrate effectiveness on real observational datasets with ambiguous interventions, latent confounding, measurement error, or domain mismatch. The claims should therefore be interpreted as strong evidence for simulated large-scale settings rather than broad real-world readiness.

3. The preliminary section explicitly assumes causal sufficiency, and the main input format includes intervention indicators appended to the data tensor. The training data also contains observational and single-node interventional samples in a 1:n ratio. This is a favorable setting for amortized discovery and should be emphasized more clearly when interpreting the strength of the empirical results.

4. The baseline set is reasonable but still incomplete for a strong top-tier benchmark claim. For example, the paper does not include broader modern scalable discovery pipelines beyond the listed methods, and several baselines are omitted from large-scale settings for runtime reasons. That is understandable, but it also makes the empirical comparison somewhat asymmetric.

5. Section 4.6 gives useful compute and memory scaling analysis, but it is informal. There is no formal proposition bounding overall runtime/memory in the full architecture, including graph stream, nor any theorem on approximation error induced by reduction.

6. The model does not enforce acyclicity during prediction and instead uses a decomposed head that empirically produces fewer cycles. This is a pragmatic design choice, but for DAG discovery it remains a conceptual weakness. The paper reports reduced cyclicity in ablations but does not deeply analyze how often invalid graphs occur in the main benchmark settings or whether downstream post-processing is required.

7. The paper shows sensitivity to OOD noise and mechanism functions, with better robustness to OOD graph structures than to OOD mechanisms/noises. This is an honest and valuable finding, but it also shows that the method’s generalization is still constrained by simulator support.

---

> ### Author Rebuttal · Authors · 2026-03-31
>
> We thank the reviewer for the insightful and perceptive review. Below we address each point in detail.
> ## Weaknesses
>
> ### W2&W3: Effectiveness on observational datasets and intervention indicators
>
> We test the model on pure observational data. Other settings are the same as in the main experiment:
>
> *n=100, |E|=400*
>
> | Method | mAP | SHD | AUC | OA |
> |--------|-----|-----|-----|----|
> | Linear | 98.3 | 39.8 | 99.9 | 99.0 |
> | NN non-add. | 86.0 | 133.6 | 98.1 | 98.8 |
> | Sigmoid | 76.4 | 155.8 | 90.9 | 88.9 |
> | Polynomial | 45.3 | 259.2 | 73.7 | 72.1 |
>
> *n=1000, |E|=2000*
> | Method | mAP | SHD | AUC | OA |
> |--------|-----|-----|-----|----|
> | Linear | 97.7 | 168.0 | 100.0 | 96.9 |
> | NN non-add. | 71.8 | 1012.8 | 97.1 | 91.9 |
> | Sigmoid | 63.1 | 1033.6 | 95.2 | 77.6 |
> | Polynomial | 28.7 | 2074.2 | 82.0 | 63.7 |
>
> From the results, we can see that the model retains high performance on observational data.
>
> ### W4: Omitted baselines and broader modern scalable discovery pipelines
>
> We agree the comparison is asymmetric, but the omissions are unavoidable due to the inability of baselines to scale; in fact, the inability of these baselines to scale is evidence of CauScale's advantage.
> Regarding broader pipelines, to the best of our knowledge, SEA (Wu et al., 2025) is the most recent amortized scalable baseline, and SDCD (Nazaret et al., ICML 2024) is a strong representative of scalable differentiable causal discovery — both are included.
> We sincerely welcome any specific suggestions from the reviewer and are happy to add further comparisons.
>
> ### W5: Formal proposition bounding
>
> **Overall runtime**: Below we provide a FLOP-level analysis of CauScale's forward pass, using the notation of the paper. We additionally let $F$ denote the FFN inner dimension. LayerNorm, dropout, and softmax are omitted as their costs are negligible relative to linear projections and attention. The total FLOP cost of CauScale is:
> $$C_\text{total} = C_\text{embed} + \sum_{b=0}^{B-1} C^{(b)} + C_\text{head}$$
> where the embed and head costs are:
> $$C_\text{embed} = 4mnd + 2n^2d, \qquad C_\text{head} \approx 4n^2d^2 + 6n^2d$$
> and the per-block cost is:
> $$C^{(b)} = 36m_b nd^2 + 8m_b n^2 d + 8m_b^2 nd + 4m_b nFd + 18n^2 d^2 + 8n^3 d + 2n^2 d + 4n^2 Fd$$
>
> **Peak inference memory**: The peak memory in the full architecture is $\Theta\bigl(mn(d{+}F)+n^2d+Hm^2\bigr)$.
>
> We will leave the detailed time and memory derivations and reduction error analysis to the revised version due to limited rebuttal space.
> ### W6：Cyclicity
>
> The model produces zero cycles across all in-distribution graph size settings. Cycles only occur in the out-of-distribution n=1000 setting, which is 2× larger than the largest graph seen during training. As all metrics we report do not require the graph to be acyclic, no special post-processing is required. Nevertheless, to further address potential concerns about acyclicity, we implement a cycle-breaking strategy to iteratively remove the edge with the lowest predicted probability. Performance is preserved or even improves after cycle-breaking. For detailed cycle-breaking results and full cyclicity statistics, see response to Reviewer grt9 Q2.
>
> ## Questions
>
> ### Q1: Scope of reported inference time
>
> The inference time reports only neural inference.
>
> ### Q2: Sensitivity to prior
>
> The model is relatively robust to inaccuracies in the prior: by design, the data stream independently processes raw observations and can compensate when the prior is noisy. To empirically validate model robustness to prior, we conduct a controlled experiment where we progressively reduce the number of samples used to compute the prior (introducing increasing inaccuracies to prior) and find that performance degrades only moderately.  For details, see response to Reviewer grt9 Q1.
>
> ### Q3: Total pretraining cost compared to AVICI or SEA
>
> As AVICI encounters out-of-memory errors on our datasets, we were unable to train it and therefore cannot report its pretraining cost. For synthetic Stage 1
> (10–100 nodes), we used SEA's publicly released checkpoint, as it was trained on
> data with the same distribution and generation code as ours, and therefore a
> direct cost comparison is not available for that stage. For synthetic Stage 2
> (150–500 nodes), CauScale takes ~22 vs. ~48 GPU-hours for SEA. For SERGIO-GRN, CauScale takes ~352 vs. ~72 GPU-hours for SEA; despite higher cost on SERGIO-GRN, CauScale substantially outperforms SEA: 71.4% vs. 5.6% (n=100) and 34.5% vs. 1.2% (n=200) in mAP. All training runs above use the same GPU settings.
>
> ### Q4: Cyclicity
>
> See response to W6.
>
> ### Q5: Performance in purely observational settings
>
> See response to W2&3.
>
> ### Q6: Performance under latent confounding
>
> We simulate latent confounding by hiding the top-$k$ nodes ranked by out-degree and evaluating on the remaining observable sub-graph. The model remains robust under moderate confounding severity. For more details, see response to Reviewer grt9 Q3.

---

> > ### Author Rebuttal · Reviewer_1vci · 2026-04-03
> >
> > Thanks for this detailed response. I have increased my score to 5

---

### Decision · Program_Chairs · 2026-04-30

**Decision:**

Accept (regular)

**Comment:**

CauScale proposes an amortized neural architecture for scalable causal discovery, introducing a reduction unit, tied attention weights, and a two-stream design (data + graph prior). It trains on 500-node graphs and infers on 1000-node graphs with 4x-13,000x speedups. Reviewers agreed that the contribution addresses a real scalability bottleneck and the empirical results are strong:

1. **Significance.** CauScale is the first amortized method to scale training to 500 nodes and inference to 1000 nodes, with substantial speedups over all baselines. The architectural solutions (reduction unit, tied attention) are coherent and well-motivated (`1vci`).

2. **Soundness.** The rebuttal substantially strengthened the paper with error estimates across all settings, AID metrics showing best causal effect estimation, noise injection robustness analysis (at most 7.7% mAP drop at α=0.8), and latent confounding experiments where CauScale outperforms all baselines even under extreme conditions (41.6% vs 25.5% next best at k=64).

3. **Originality.** The two-stream design and progressive reduction are novel architectural contributions, properly contextualized as engineering advances rather than new identifiability theory (`1vci`).

Scores were 4, 4, 5, 5 (mean 4.5). `1vci` (highest confidence, 4) scored 5 after the rebuttal. `grt9` (score 4) raised concerns about sensitivity to the precision matrix prior; in the discussion phase, `grt9` acknowledged that the noise injection experiment addressed this concern but maintained reservations about cyclicity at scale and OOD degradation, stating "I will update my score" without specifying a final number. I have reviewed the authors' AC confidential comment regarding `grt9`'s interpretation of the sensitivity analysis and find the authors' clarification (graph size was held fixed; only sample count varied) persuasive. The paper is recommended for acceptance.

**Revision requirements:** (1) Cyclicity statistics and cycle-breaking impact must appear in the main text — at n=1000 polynomial, 103 edges are removed per graph on average, which is non-trivial for a scaling-focused method. (2) Error estimates and AID results from the rebuttal must be included in the appendix as committed. (3) Related work must discuss Granger causal methods and explicitly state the method is amortized, as agreed with `u8Ys`. (4) Synthetic-only evaluation should be clearly acknowledged as a limitation.